# Information-Theoretic State Variable Selection for Reinforcement Learning

## Abstract

Identifying the most suitable variables to represent the state is a fundamental challenge in Reinforcement Learning (RL). These variables must efficiently capture the information necessary for making optimal decisions. In order to address this problem, in this paper, we introduce the Transfer Entropy Redundancy Criterion (TERC), an information-theoretic criterion, which determines if there is *entropy transferred* from state variables to actions during training. We define an algorithm based on TERC that provably excludes variables from the state that do not affect the agent's policy during learning, resulting in more efficient inference. Our approach is policy-dependent, making it agnostic to the underlying learning paradigm. Consequently, we use our method to enhance efficiency across three different algorithm classes (represented by tabular Q-learning, Actor-Critic, and Proximal Policy Optimization (PPO)) in a variety of environments. Furthermore, to highlight the differences between the proposed methodology and the current state-of-the-art feature selection approaches, we present a series of controlled experiments on synthetic data, before generalizing to real-world decision-making tasks. We also introduce a representation of the problem that compactly captures the transfer of information from state variables to actions as Bayesian networks.

## 1 Introduction

The choice of an appropriate state representation remains a key design challenge of every Reinforcement Learning (RL) system. This process involves finding simplified representations of the information required to learn optimal policies. In order to achieve this, various techniques have been proposed (Finn et al., 2016; Laskin et al., 2020; Gelada et al., 2019; Stooke et al., 2021). Nevertheless, several issues can be identified with these approaches. Firstly, these methods fail to provide a mechanism by which the state's dimensionality can be reduced. Consequently, at the time of deployment, most RL systems process unnecessarily large amounts of data. Second, they are used in a black-box fashion, providing no insight into the final form of the simplified representation. Finally, they fail to provide a solution for determining the appropriate state history lengths when temporally extended states are required, as demonstrated in Mnih et al. (2015); Pan et al. (2017). As a result of these shortcomings, most RL state design applications fall into the realm of intuition-guided heuristics (Dean & Givan, 1997; Ortiz et al., 2018; Reda et al., 2020; Liu et al., 2020). The use of such *ad hoc* methods results in state representations that incorporate non-informative variables, which might increase the training duration due to Bellman's curse of dimensionality (Bellman & Kalaba, 1959). They may also lead to state representations that lack the required information, preventing the agent from learning optimal policies.

In order to provide an intuition of the problem, let us consider the classic cart pole environment[1]. Imagine a cart that can move left or right along a track. On top of this cart, there is a pole standing upright. The goal is to keep the pole balanced and prevent it from falling over. You can do this by moving the cart left or

---

[1]Many implementations of this environment exist, like that implemented in the now publicly maintained Gym/Gymnasium framework (`https://gymnasium.farama.org/environments/classic_control/cart_pole/`). In the evaluation of TERC presented in a later section, we use the original implementation by OpenAI (Brockman et al., 2016).

right to keep the pole centered. The challenge is to figure out the best way to move the cart to keep the pole balanced for as long as possible. Typically, at time $t$, this environment is characterized by a 4-dimensional set of variables $s^t = [x^t, \theta^t_{pole}, \dot{x}^t, \dot{\theta}^t_{pole}]$, where $x$ indicates the cart's $x$-position, $\theta^t_{pole}$ describes the angle of the pole, and $\dot{()}$ denotes the derivative with respect to time. Let us suppose that an additional random variable $v_{rand}$, which is independent of the physical system under consideration, is added to the environment. The resulting state will be $s^t_{extended} = [x^t, \theta^t_{pole}, \dot{x}^t, \dot{\theta}^t_{pole}, v_{rand}]$. $v_{rand}$ is not informative for the problem at hand and a method to define optimal representations should be able to exclude it. Failure to remove such variables will lead to unnecessary computations once deployed[2]. Moreover, uninformative variables can impede the learning process of iterative decision-making systems (Grooten et al., 2023).

To address these issues, we propose the Transfer Entropy Redundancy Criterion (TERC) for the selection of the minimal set of the state variables. TERC is a criterion that allows us to determine *whether state variables transfer entropy to actions during training* (Schreiber, 2000). More specifically, TERC is based on the quantification of the reduction in uncertainty of the realizations of the policy when considering the set of state variables with and without the variable under consideration. This makes TERC a policy-dependent method, while being agnostic to the underlying learning algorithm. If this value is bigger than zero, the actions are said to depend on this state variable, and TERC is verified. Under these circumstances, the state variable is considered informative and included in our representation. Once all the informative state variables have been identified, our agent can be re-trained on the lightweight state, leading to greater efficiency at the time of deployment. We will provide the reader with a formal definition of TERC in Section 5.2[3].

In practical terms, we will introduce a set of methods that -in conjunction with the satisfaction of an assumption- will allow us to derive the minimal and optimal set of variable for state representation in presence of redundant and synergistic relationships. Even when this assumption does not hold, our method still identifies an optimal, though not necessarily minimal, representation. This set is constructed starting from an empty set and adding variables to it by verifying TERC. Indeed, the core principle of our methodology is to include only the variables in the state representation that are informative. We will show that TERC can achieve this even in the presence of perfectly redundant state variables (Frye et al., 2020; Kumar et al., 2020). Consequently, we will show that it is possible to obtain an optimal subset whose realizations reduce the entropy of the actions identically to the original set of all potential state variables. Additionally, we will demonstrate that this method can serve a dual purpose: not only does it help in omitting uninformative variables from our representation, but it also facilitates investigating how variable importance varies as the agent trains, thereby enhancing the overall interpretability of the learning process.

In addition to discussing the theoretical basis of TERC, to further validate our approach from a practical point of view, we will present an extensive experimental evaluation through a selection of novel and existing environments. We will firstly apply TERC to purely synthetic data, considering different complex relationships between the given state variables. We will then introduce the 'Secret Key Game', a novel environment in which an agent is tasked with learning a secret message from a state composed of not only 'secret keys' (from which the secret message is calculated), but also superfluous 'decoy keys'. The calculation of the secret message is based on a classic method for secure multiparty communication, as described in (Shamir, 1979; Blakley, 1979). We will then consider a set of physics-based examples. We will consider three very popular Gym environments: Cart Pole, Lunar Lander, and Pendulum. Finally, we will illustrate the versatility of TERC, applying it to strategic sequential games, in which the temporal dimension plays a key role. In particular, we will focus on the problem of learning optimal state history lengths when playing against a Tit-For-N-Tats (TFNT) opponent in the Iterated Prisoner's Dilemma.

**Summary of contributions.** The contributions of this paper can be summarized as follows:

- We propose TERC, an effective criterion for the selection of state variables in reinforcement learning, discussing its theoretical foundations in detail.

---

[2]Indeed, this is of critical importance given the increasing energy footprint of today's machine learning systems (Patterson et al., 2022).

[3]The code for the derivation of the measures at the basis of TERC is available at this URL: [*We plan to release the TERC library and the code used for all the simulations presented in this manuscript upon its acceptance*].

- We analyze potential issues arising from the presence of perfect conditionally redundant variables in the state and we introduce algorithmic solutions to deal with them.

- We corroborate our theoretical results experimentally by means of an extensive evaluation considering a variety of RL environments.

- Finally, we discuss the extent to which TERC enhances the interpretability of RL systems and its practical applications.

## 2 Related Work

As discussed in the introduction, the method presented in this paper is related to the field of feature selection in machine learning. The proposed solution is then applied to the problem of state representation in RL. Consequently, we will first review the state of the art in the area of feature selection methods before discussing the relevant work on state representation learning.

**Feature selection.** Our objective is to devise a measure that assesses how informative state variables are given an agent's actions. This is conceptually similar to the idea of 'feature importance' in the field of feature selection (Ribeiro et al., 2016; Shrikumar et al., 2017; Plumb et al., 2018; Sundararajan & Najmi, 2020; Chen et al., 2020). The exact definition of 'feature importance' is an open problem *per se* (Catav et al., 2021; Janssen et al., 2023). However, it is generally agreed that its practical objective is to rank the input variables of a machine learning model based on their ability to predict the model output. For example, Shapley values, originally introduced in a game-theoretic context for allocating resources in cooperative systems (Shapley, 1953), have been repurposed as indicators of feature importance. One of the first examples was Multi-perturbation Shapley value Analysis (MSA) (Keinan et al., 2004; Cohen et al., 2007), which entails considering features as agents which cooperate to predict model outcomes. The Shapley Additive Explanation (Lundberg & Lee, 2017) and Shapely Additive Global Explanation (Lundberg & Lee, 2017) models expanded on this, allowing the ranking of input features both globally (predicting all model outputs) and locally (predicting a single model output). Despite the success of Shapley value-based methods (Apley & Zhu, 2020; Covert et al., 2020; Kwon & Zou, 2022), both theoretical and experimental findings have shown that the presence of redundant information can invalidate these methods' results (Frye et al., 2020; Kumar et al., 2020). To deal with these issues, in this work we draw inspiration from information-theoretic filter feature selection methods (Battiti, 1994; Peng et al., 2005; Brown et al., 2012; Gao et al., 2016; Chen et al., 2018). Rather than ranking variables by feature importance, we propose the use of a 'target set' that we populate with variables that lead to desirable properties. This is analogous to what is proposed by Peng et al. (2005); Gao et al. (2016); Chen et al. (2018), except here we favorably derive the subset size, rather than requiring it as an arbitrary hyperparameter. Brown et al. (2012) proposed a method that did not suffer from such drawbacks, which has inspired subsequent works (Covert et al., 2023; Wollstadt et al., 2023). However, they fail to highlight optimal feature sets in some cases. Their methods add new features to the selected subset if they enhance the overall correlation with the target. This approach overlooks features that only become informative when considered in combination. Furthermore, such methods add features that maximize the correlation function. This function must be computed for each feature at every step, meaning the complexity of these techniques fails to scale linearly in time with respect to the number of features (Borboudakis & Tsamardinos, 2019; Tsamardinos et al., 2019; Hao et al., 2021). Our method has the ability to evaluate the effects of complex relationships, such as redundancies and synergistic combinations, while remaining linear in time with respect to the number of features.

**Unsupervised derivation of state representations in reinforcement learning.** The Infomax principle has been used as the basis for several unsupervised representation learning techniques; it involves the maximization of the mutual information (MI) between network inputs and appropriately constrained outputs (Linsker, 1988; Bell & Sejnowski, 1995). The methods relying on the Infomax principle have been successfully used to learn representations of natural language (Devlin et al., 2019), videos (Sun et al., 2019), images (Hjelm et al., 2019; Bachman et al., 2021), and RL states (Finn et al., 2016; Laskin et al., 2020). For high-dimensional and continuous input/output spaces, the associated MI is not computable, and, for this

reason, the application of the Infomax principle is not practically possible (Song & Ermon, 2020). To address this issue, a variety of methods for estimating MI have been developed (van den Oord et al., 2019; Belghazi et al., 2018; Poole et al., 2019). The problem of defining a compact state representation has been studied primarily as an abstraction problem (James C. Bean, 1987; Lesort et al., 2018). The goal of state abstraction is to reduce the size of the associated state space by unifying the representation of the areas of the space that provide indistinguishable information. Examples of such techniques include bisimulation (Givan et al., 2003), homomorphism (Ravindran & Barto, 2003), utile distinction (Mccallum & Ballard, 1996), and policy irrelevance (Jong & Stone, 2005). These methods most commonly leverage MI-based abstraction principles (such as Infomax) to redefine the distributions associated with each of the original state variables, leading to new representations that preserve MI between sequential states, actions, or combinations thereof (Schwarzer et al., 2020); while concurrently disregarding irrelevant and redundant information. In our study, we take a different approach, focusing on the conditional relationships between state variables and actions during the learning phase. This allows us to eliminate variables that are not informative with respect to agent training. Upon completion of this process, we are not required to consider the entire set of variables, in contrast to the methods previously discussed. In other words, our solution derives a compact state representation with fewer variables. Once deployed, this reduced subset requires less computation for measuring, processing, and storing. Additionally, in the case of applications that require a temporally-extended state representation (e.g., a sequence of frames for an arcade game), our method is able to derive the optimal state history length.

## 3 Background and Notation

In the following, calligraphic symbols (e.g., $\mathcal{X}$) indicate sets, capital letters (e.g., $X$) are used to represent random variables, whereas their realizations are denoted with lower-case letters (e.g., $x$). We do not make assumptions in terms of the distributions associated with the random variables unless otherwise stated. We will use Shannon's entropy, denoted as $H(\cdot)$ (Shannon, 1948), to quantify uncertainty: an increase in $H(\cdot)$ signifies a rise in uncertainty. In addition, the term 'dependence' is used to describe how the observation of one variable or group of variables reduces the entropy of another variable or group of variables.

### 3.1 Reinforcement Learning

In this section, we briefly introduce a set of definitions concerning Markov Decision Processes (MDPs) and other basic RL concepts, including the notation we use throughout this paper. Finally, we explain how we represent states as sets of random variables, which will constitute the basis of our information-theoretic analysis.

An MDP is characterized by a tuple $\mathcal{M} = (\mathcal{S}, \mathcal{A}, \mathrm{R}, \mathrm{T})$, which consists of the set of all possible states $\mathcal{S}$, the set of all possible actions $\mathcal{A}$, a reward function R, and a transition function T. The agent, i.e., the decision-maker, interacts with its environment by taking an action $a^t(s^t) \in \mathcal{A}$, which is dependent on its current state $s^t \in \mathcal{S}$. This leads to distributions over the possible subsequent rewards $r^{t+1} = \mathrm{R}(s^t, a^t)$ and states $s^{t+1} = \mathrm{T}(s^t, a^t)$. The agent continues taking actions until it reaches a terminal state at time $T$. The resulting time-ordered list of states and actions is referred to as a trajectory $(s^1, a^1, s^2, a^2, \ldots, s^T, a^T)$, where a state at time $t$ is a vector such that $s^t = [x_1^t, x_2^t...x_N^t]$. The aim of RL is to allow the agent to learn, through repeated experience, a policy $\pi(a^t|s^t)$, such that it maximizes the total cumulative reward of each trajectory $J(\pi) = \mathbb{E}_\pi[\sum_{t=0}^{\infty} \gamma^t r^t]$, where $\gamma \in [0,1]$ is a discount factor. A variety of RL algorithms have been proposed over the past few decades (Sutton & Barto, 2018). It is worth noting that the method discussed below is compatible with all the existing RL approaches presented in the literature. In the following, we will demonstrate its applicability for a set of algorithms, which are representative of different classes of RL approaches, namely Q-learning (Watkins & Dayan, 1992)), Actor-Critic (Konda & Tsitsiklis, 1999), and Proximal Policy Optimization (PPO) (Schulman et al., 2017).

After convergence, we sample the values (e.g., $x_i^t$) of each variable (e.g., $X_i$) from the training trajectories, to derive the approximate distribution (e.g., $p_{X_i}$). We also sample from $s^t = [x_1^t, x_2^t, \ldots, x_N^t]$ to derive the set of random variables $\mathcal{X} = \{X_1, X_2, \ldots, X_N\}$. In order to derive the random variables associated with the agent's actions, we follow an identical sampling process, leading to $A$. We also sample from

$s^t = [x_1^t, x_2^t, \ldots, x_N^t]$ to derive the distribution set of random variables $\mathcal{X} = \{X_1, X_2, \ldots, X_N\}$, such that $p(\mathcal{X} = s) = 1/T \sum_{i=1}^T p(\mathcal{X}^t = s)$. In order to derive the random variables associated with the agent's actions, we follow an identical sampling process, leading to $A$, where $p(A = a) = 1/T \sum_{i=1}^T p(A^t = a)$.

## 3.2   Transfer Entropy

Transfer entropy (TE) can be considered a nonlinear quantification of Granger causality, as applied to time series. It measures how the knowledge of variable $Y^{t-1}$ reduces the uncertainty about the variable $X^t$ when compared to $X^{t-1}$ (Granger, 1969; Schreiber, 2000; Seitzer et al., 2021). Similarly, our method relies on the ability to quantify how the knowledge of the state variables reduces the uncertainty of the values of actions when compared with other state variables. In this section, we first formally define TE before reviewing methods for estimating and graphically representing this quantity. We adopt the following definition of TE (Schreiber, 2000):

$$TE_{Y \to X} = \mathbb{E}_{p_{X,Y}(x^t, x^{t-1}, y^{t-1})} \log \frac{p_{X,Y}(x^t | x^{t-1}, y^{t-1})}{p_{X,Y}(x^t | x^{t-1})} = H(X^t | X^{t-1}) - H(X^t | X^{t-1}, Y^{t-1}) \tag{1}$$

where $\mathbb{E}_{p_{X,Y}(x^t, x^{t-1}, y^{t-1})}$ is the expected value over all possible realizations of $x^t$, $x^{t-1}$, and $y^t$ and $H$ is the Shannon's entropy. In many real-world environments, state and/or action spaces are usually highly dimensional and continuous. In this case, the calculation of TE is intractable; therefore, we estimate it using a function approximation method. In order to do so, we first estimate MI and from it calculate TE via the following relationship $TE_{Y \to X} = I(X^t; X^{t-1}, Y^{t-1}) - I(X^t; X^{t-1})$.

A variety of methods have been derived in order to estimate MI (Moon et al., 1995; Paninski, 2003; Belghazi et al., 2018; van den Oord et al., 2019; Poole et al., 2019). In this work, we adopt the solution proposed by Belghazi et al. (2018) because of its general nature and applicability to continuous, discrete, and multivariate domains. Moreover, for interpretability, we represent the TE between variables using Bayesian networks, as defined in Pearl (1986).

## 3.3   Constrained Perfect Multivariate Conditional Redundancy

In this section, we introduce Constrained Perfect Multivariate Conditional Redundancy (CPMCR). We will use this extended formulation to overcome the redundancy problem when selecting state variables (Shapley, 1953; Frye et al., 2020; Kumar et al., 2020). CPMCR will be defined as a condition that, if true, signifies the existence of two subsets $(\mathcal{P}, \mathcal{P}' \in \mathscr{P}(\mathcal{X}) : \mathcal{P} \neq \mathcal{P})$ that convey identical information about the target. The underlying principle is to use this condition to verify whether such subsets exist, before including the smallest of these in our final state representation. By doing so, we will be able to minimize the cardinality of the final state representation.

To begin, we note that if there are variables in the set $\mathcal{X}$ upon which $Y$ has no dependence, adding them to either subset $\mathcal{P}$ or subset $\mathcal{P}'$ has no impact on their information-theoretic properties. However, our goal is to define CPMCR without including these uninformative variables. Therefore, we first introduce the following condition:

$$\psi_{\mathcal{P}'} = (\forall P' \in \mathcal{P}' : H(Y | \mathcal{X}_{\backslash(\mathcal{P}, \mathcal{P}')} \cup \mathcal{P}') < H(Y | \mathcal{X}_{\backslash(\mathcal{P}, \mathcal{P}')} \cup \mathcal{P}'_{\backslash P'})), \tag{2}$$

where $P'$ is an element of $\mathcal{P}'$. This condition can be interpreted as follows: adding the full subset $\mathcal{P}'$ back to the set $\mathcal{X}_{\backslash(\mathcal{P}, \mathcal{P}')}$ leads to a greater reduction in uncertainty about $Y$ than adding an incomplete subset of $\mathcal{P}'$. Likewise, a similar statement can be made about $\mathcal{P}$, which we denote as $\psi_{\mathcal{P}}$. We are now in a position to introduce the formal definition of CPMCR. We denote this quantity as $\Psi(Y | \mathcal{X})$ and define it as follows:

$$\begin{aligned}
\Psi(Y | \mathcal{X}) = (&\exists \mathcal{P} \in \mathscr{P}(\mathcal{X}), \exists \mathcal{P}' \in \mathscr{P}(\mathcal{X}_{\backslash \mathcal{P}}) : \\
& H(Y | \mathcal{X}) = H(Y | \mathcal{X}_{\backslash \mathcal{P}'}) = H(Y | \mathcal{X}_{\backslash \mathcal{P}}) < H(Y | \mathcal{X}_{\backslash(\mathcal{P}, \mathcal{P}')}) \quad \& \\
& \psi_{\mathcal{P}'} \quad \& \\
& \psi_{\mathcal{P}}).
\end{aligned} \tag{3}$$

The condition above, if verified, indicates the existence of two subsets that reduce the uncertainty of the target by the same amount. Given these conditions, many existing methods for calculating feature importance fail to produce correct results (Breiman, 2001; Debeer & Strobl, 2020) as theoretically and empirically shown by Frye et al. (2020); Kumar et al. (2020). For instance, let us assume there are two state variables that provide perfectly conditionally redundant information regarding an agent's actions. In our optimal set of state variables, only one of these two variables is necessary. Existing techniques struggle to deal with this type of situation; some methods would include both variables (Catav et al., 2021; Janssen et al., 2022), while others would include neither (Debeer & Strobl, 2020).

### 3.4 Synergy

Another fundamental multivariate concept, which is not often discussed in the feature selection literature (a rare example is Wollstadt et al. (2023)) is *synergy*. Synergy characterizes the additional information obtained by evaluating variables collectively rather than individually. Conceptually, it can be viewed as the antithesis of redundancy. Intriguingly, synergy is inherently conditional. This stems from the observation that when variables combine to provide additional information, that information must pertain to a specific target variable. More formally, synergy can be defined as follows (Anastassiou, 2007):

$$Syn(Z; \mathcal{X}) = I(Z; \mathcal{X}) - \sum_{i=1}^{N} I(Z; X_i). \tag{4}$$

The characteristics of this relationship can be illustrated by means of the XOR function. Consider two binary string variables, $X_1$ and $X_2$, with $Z$ being their XOR output. In this scenario, $X_1$ and $Z$, as well as $X_2$ and $Z$, are uncorrelated ($I(X_i; Z) = I(X_j; Z) = 0$), but together, $X_1$ and $X_2$ fully describe $Z$ ($I(X_1, X_2; Z) = H(X)$) (Guyon & Elisseeff, 2003; Williams & Beer, 2010). Such relationships demand careful consideration in feature selection, as correlation-based measures (like those in Brown et al. (2012); Covert et al. (2023)) would deem these features unimportant. Methods to address this challenge exist, but they typically involve computationally intensive searches of subset permutations. Some approaches limit this computational burden through gradient descent-guided subset search (Covert et al., 2020; Chen et al., 2020; Yamada et al., 2020; Catav et al., 2021; Covert et al., 2023). However, these fail to guarantee optimality. Meanwhile, ablation-based methods (Breiman, 2001; Debeer & Strobl, 2020) naturally address these issues but tend to exclude redundant variables.

## 4 Problem Statement

We now introduce the problem statement in a formal way. Given the set of all observable state variables, $\mathcal{X} = \{X_1, X_2...X_N\}$, we aim to obtain the smallest possible subset $\mathcal{X}_* \subseteq \mathcal{X}$ from which RL agents can learn optimal policies, such that $|\mathcal{X}_*| \leq |\mathcal{X}|$.

To achieve this, we first train an agent until convergence using all the observable variables as the state $s^t = [x_1^t, x_2^t...x_N^t]$, where $s^t \in \mathcal{S}$ and $x_i$ is a realization of $X_i \in \mathcal{X}$. We then isolate the actions $A$ and state variable distributions $\mathcal{X}$ (see Section 3.1 for the definition of $A$ and $\mathcal{X}$). Subsequently, we identify the state variables upon which the agent's actions exhibit zero dependence. Therefore, these variables are irrelevant during the training of the agent and can be excluded from the state representation. The remaining set of variables $\mathcal{X}_*$ are a minimal subset $\mathcal{X}_* \subseteq \mathcal{X}$, which, when observed, reduce the entropy of $A$ identically to the complete set of variables $\mathcal{X}$ (Brown et al., 2012). Formally, this set is defined as follows[4]:

$$\mathcal{X}_* \in \{\mathcal{P} \in \mathscr{P}(\mathcal{X}) : (|\mathcal{P}| = \min_{H(A|\mathcal{P})=H(A|\mathcal{X})} |\mathcal{P}|) \quad \& \quad (H(A|\mathcal{P}) = H(A|\mathcal{X}))\}. \tag{5}$$

Because the state variables specified in Equation 5 convey the same informational as the full set, they share the convergence guarantees established by Theorem 4(1) in (Li et al., 2006). Additionally, diminishing the dimensionality of the state space results in a proportional reduction of the regret bounds (Yang & Wang,

---

[4]It is worth noting that this set is defined as a member of a set of subsets, due to the existence of multiple representations that satisfy the conditions introduced in the problem statement.

2020). Our aim is to develop an approach to derive $\mathcal{X}_*$, which can the be used as the new state representation for agent training, such that $s_*^t = [x_1^t, x_2^t...x_M^t]$, where $s_*^t \in \mathcal{S}_*$ and $x_i^t$ is the realization of $X_i \in \mathcal{X}_*$.

## 5  Approach

In this section, we will present the design of TERC, first discussing a naïve solution and then examining key aspects of the problem, namely perfect conditional redundancy and synergy.

### 5.1  Overview

We aim to address the problem outlined in Section 4 through the following steps:

1. We introduce the mathematical details of TERC, the criterion used to determine whether actions depend on state variables (see Section 5.2).

2. We start from a straightforward method based on TERC that leads to the derivation of an appropriate representation of $\mathcal{X}_*$. However, this method is valid only under the unrealistic assumption of absence of CPMCR; for this reason, we refer to it as a naïve solution (see Section 5.3).

3. Finally, we present a practical yet optimal method based on TERC that can be applied also in presence of CPMCR under a weak assumption (see Section 5.4).

### 5.2  Design of the Transfer Entropy Redundancy Criterion (TERC)

We now introduce the mathematical details of the Transfer Entropy Redundancy Criterion (TERC). TERC is based on the concept of transfer entropy (TE), as we are interested in how variables *influence* actions using Granger's interpretation (Granger, 1969). Specifically, TERC quantifies the reduction in uncertainty associated with realizations of $A$ when considering the set $\mathcal{X}$ with and without a given variable. If this value is bigger than zero, the actions are said to depend on this state variable, and TERC is verified. More formally, we define TERC as follows:

$$\Phi_{X_i;\mathcal{X}\to A} = H(A|\mathcal{X}_{\setminus X_i}) - H(A|\mathcal{X}) > 0. \tag{6}$$

$\Phi_{X_i;\mathcal{X}\to A}$ describes how actions depend conditionally on state variables; therefore, we graphically represent these directed dependencies as Bayesian networks. For a full description of how to estimate the measure defined in Equation 6, refer to Algorithm 2 in Appendix A.

$H(A|\mathcal{X}_{\setminus X_i}) - H(A|\mathcal{X})$ quantifies the reduction in entropy of the realizations of $A$, when variable $X_i$ is removed from the set $\mathcal{X}$. Positive values of this quantity indicate the actions are dependent on $X_i$ and, therefore, they should be maintained in the state representation. In this case, we describe $X_i$ as satisfying TERC. Conversely, if there is no reduction in the entropy of the values of $A$, then $\Phi_{X_i;\mathcal{X}\to A} = 0$[5] and the variable does not satisfy TERC. In this case, by definition, we can say that the information provided by variable $X_i$ is either irrelevant or redundant and removing $X_i$ from the state should lead to a policy that can perform inference more efficiently. It follows that we should only add variables to $\mathcal{X}_*$, the subset of maximal information and minimal cardinality, if they satisfy TERC ($\Phi_{X_i;\mathcal{X}\to A} > 0$).

### 5.3  A Naïve Solution

We now describe a simple application of TERC, our previously defined criterion. To begin, we instantiate an empty set $\mathcal{X}_\Phi$. We then populate this set via the simultaneous addition of variables if their removal from set $\mathcal{X}$ increases the conditional entropy of $A$. We write this more formally as:

$$\mathcal{X}_\Phi = \{X_i \in \mathcal{X} : \Phi_{X_i;\mathcal{X}\to A} > 0\}. \tag{7}$$

---

[5]This is due to the non-negativity of the measure defined in Equation 6, a property that we prove in Appendix B.

---

**Algorithm 1** A Simple State Variable Selection Method Based on TERC

---

**Input**: State and action variables generated by sampling from learning trajectories: $\mathcal{X}$ and $A$ respectively (see Section 3.1).
**Output**: The smallest subset of $\mathcal{X}$ that still fully describes the agents actions: $\mathcal{X}_{A_1}$

1: Initialize $\mathcal{X}_{A_1} = \{\}$
2: **for** $i = 1$ to $N$ **do**
3:    **if** $\Phi_{X_i; \mathcal{X} \to A} = 0$ **then**
4:       $\mathcal{X} = \mathcal{X} \backslash \{X_i\}$
5:    **else**
6:       $\mathcal{X}_{A_1} = \mathcal{X}_{A_1} \cup \{X_i\}$
7:    **end**
8: **end for**
9: **return** $\mathcal{X}_{A_1}$

---

However, in case of CPMCR, the subset $\mathcal{X}_\Phi$ does not satisfy $H(A|\mathcal{X}_\Phi) = H(A|\mathcal{X})$. We formally write this by means of the following lemma:

**Lemma 1.** *Given there exists a case of CPMCR in the set $\mathcal{X}$, such that $\Psi(A|\mathcal{X})$, the following result holds: $\Psi(A|\mathcal{X}) \Rightarrow H(A|\mathcal{X}) < H(A|\mathcal{X}_\Phi)$.*

*Proof.* Refer to Appendix C.

In Lemma 1, we show that, if $\Psi(A|\mathcal{X})$ is verified, $H(A|\mathcal{X}) < H(A|\mathcal{X}_\Phi)$ and, therefore, $\mathcal{X}_\Phi \neq \mathcal{X}_*$. On the contrary, by assuming that there are no cases of CPMCR in $\mathcal{X}$, we can prove the following:

**Theorem 1.** *Given there are no cases of CPMCR in the set $\mathcal{X}$, the set defined in Equation 7 satisfies the following: $\neg\Psi(A|\mathcal{X}) \Rightarrow \mathcal{X}_* = \mathcal{X}_\Phi$.*

*Proof.* See Appendix D.

However, $\neg\Psi(A|\mathcal{X})$ is not always satisfied.

### 5.4 TERC in Practice

In the preceding section, we presented a naive application of our criterion TERC, showing that CPMCR cases can prevent us from achieving the goal outlined in the problem statement. In this section, we address this issue by presenting an algorithm—valid under a weak assumption—that applies TERC in a computationally efficient manner. To begin, we motivate the need for our condition before formally introducing it and our algorithm.

Instead of adding features to our selected set simultaneously, suppose we iterate through them randomly and add them one at a time while re-verifying TERC. As a result, if two variables are perfectly redundant, the first one encountered will be removed. Upon re-verifying TERC for the second redundant variable, the removal of the first renders the second non-redundant. Consequently, we negate the issue described in the preceding section. However, if rather than consider two redundant variables we consider two redundant subsets, we have no mechanism that ensures we encounter and remove the smaller of the two subsets. Therefore, we can only be sure that we include the minimum number of variables if both redundant subsets are of equal size. This observation leads us to define the following condition:

**Condition 1.** *Let $\psi_\mathcal{P}$ be as defined in Equation 2 and let $\mathcal{P} \in \mathscr{P}(\mathcal{X})$. We define the condition as follows*

$$
\begin{aligned}
C_1 = (\forall \mathcal{P} \in \mathscr{P}(\mathcal{X}), \nexists \mathcal{P}' \in \mathscr{P}(\mathcal{X}) : &|\mathcal{P}| \neq |\mathcal{P}'| \quad \& \\
&H(A|\mathcal{X}) = H(A|\mathcal{X}_{\backslash \mathcal{P}'}) = H(A|\mathcal{X}_{\backslash \mathcal{P}}) < H(A|\mathcal{X}_{\backslash(\mathcal{P}, \mathcal{P}')}) \quad \& \\
&\psi_\mathcal{P} \quad \& \\
&\psi_{\mathcal{P}'}).
\end{aligned}
\tag{8}
$$

We now more formally describe the approach, which guarantees the derivation of $\mathcal{X}_*$, provided condition $C_1$ is true. Because the subset derived using this approach requires the use of Algorithm 1 ($A_1$), it will be labeled $\mathcal{X}_{A_1}$. We write the three steps involved in this approach as follows:

1. Generate trajectories by training an agent using all observable variables as the state $s^t = [x_1^t, x_2^t ... x_N^t]$, where $s^t \in \mathcal{S}$ and $x_i^t$ is a realization of $X_i \in \mathcal{X}$.
2. Iterate through variables $X_i \in \mathcal{X}$ and remove variables from $\mathcal{X}$ that satisfy $\Phi_{X_i;\mathcal{X}\to A} = 0$, otherwise add them to $\mathcal{X}_{A_1}$.
3. Use this newly designed state for agent training, such that $s_{A_1}^t = [x_1^t, x_2^t ... x_N^t]$, where $s_{A_1}^t \in \mathcal{S}$ and $x_i^t$ is a realization of $X_i \in \mathcal{X}_{A_1}$.

Notably, the complexity of Algorithm 1 scales linearly in time with respect to the number of features. This makes it preferable to pre-existing techniques that include or exclude features that maximize some function based on the feature added at each step (Brown et al., 2012; Wookey & Konidaris, 2015; Gao et al., 2016; Borboudakis & Tsamardinos, 2019; Tsamardinos et al., 2019; Covert et al., 2023; Bonetti et al., 2023). We now introduce theoretical guarantees for this method.

**Theorem 2.** *Let the subset $\mathcal{X}_{A_1}$ be generated using Algorithm 1 and let $C_1$ be as is defined in Equation 8, we can write the following: $C_1 \Rightarrow \mathcal{X}_* = \mathcal{X}_{A_1}$.*

*Proof.* See Appendix E.

In practice, this condition is satisfied in many scenarios, including all of the cases studied in the upcoming experimental evaluation.

### 5.5 Methods for Approximating State Variable Inclusion Conditions

Since we neurally estimate $\Phi_{X_i;\mathcal{X}\to A}$ instead of computing it directly, the estimates are susceptible to noise, which may lead to the inclusion of uninformative variables in our target set. Similarly to Wollstadt et al. (2023), to approximate the condition $\Phi_{X_i;\mathcal{X}\to A} = 0$ for fluctuating estimates, we adopt a null model to which the values of $\Phi_{X_i;\mathcal{X}\to A}$ can be compared. This is done by introducing a random variable, $NM$, once training is complete. In theory, the actions should have no dependence on $NM$; therefore, $\Phi_{NM;\mathcal{X}\to A}$ should approach zero under all circumstances. Hence, variables that transfer entropy to actions are expected to deviate from the null model results. We assume normality for both the distribution of the null model and the variables in $\mathcal{X}$. We then use the methods outlined in (Neyman & Pearson, 1933) to determine which $X_i \in \mathcal{X}$ have values of $\Phi_{X_i;\mathcal{X}\to A}$ with a 95% chance of falling outside the range described by the null model. These variables are considered to show a statistically significant deviation from the null model, and therefore satisfy $\Phi_{X_i;\mathcal{X}\to A} > 0$.

In the upcoming experimental evaluation, we will present the upper bound of the null model's 95% confidence interval as a red dashed line. State variables whose lower bound exceeds this dashed line are considered to satisfy TERC, and are included in our representation. Otherwise, they are excluded.

## 6 Experimental Evaluation

We now discuss the experimental evaluation of TERC. We first outline the experimental settings and the baselines used for comparison. We then conduct an extensive experimental evaluation, considering synthetic data and RL games of increasing complexity. Finally, we examine the process of identifying an optimal state history length adopting the Iterated Prisoner's Dilemma as case study.

### 6.1 General Experimental Settings

The calculation of $\Phi_{X_i;\mathcal{X}\to A}$ is derived over 10 runs. The results of the experiments using RL environments are obtained over 5 runs. We indicate the null model using a red dotted line. Hyperparameter tuning is conducted via a simple grid search. We adopt $\pm 95\%$ confidence intervals for both the measure-estimation plots and agent performance curves.

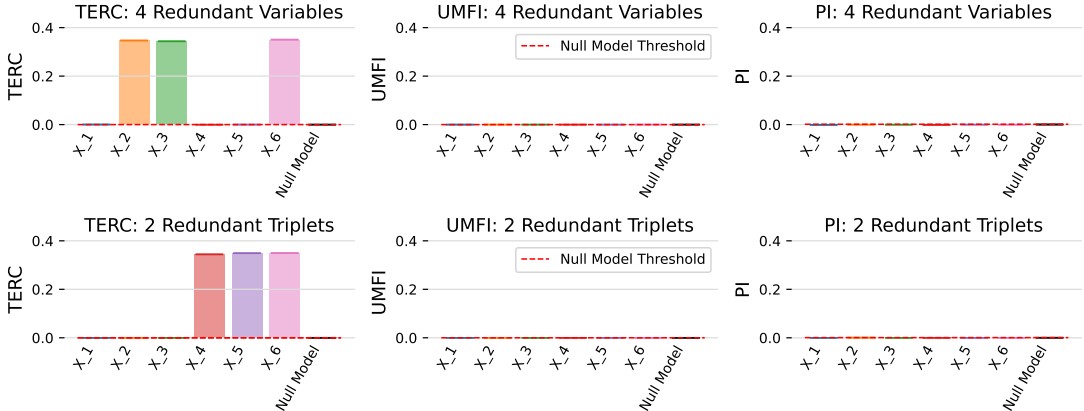

Figure 1: In this graph we illustrate TERC's effectiveness in dealing with complex redundancies and synergies. We plot the values for TERC, PI and UMFI for the *Four Redundant Variables* and the *Two Redundant Triplets* datasets.

## 6.2 Baselines

In this section, we introduce two existing methods for feature selection that we will use as baselines for our experiments. For both techniques, we will again adopt a null model when systematically deciding whether or not to include a variable in our final set (see Section 5.5). The first baseline we adopt is the Ultra Marginal Feature Importance (UMFI) algorithm with optimal transport as described by Janssen et al. (2023). This state-of-the-art method was selected since it is the only approach in the existing literature that aims to resolve redundancies with, at the same time, a temporal complexity that scales linearly with respect to the number of features. The second baseline is the widely adopted Permutation Importance (PI) algorithm, initially described in (Breiman, 2001), which is also considered computationally efficient. For UMFI and PI, similarly to TERC, hyperparameter tuning is performed via a simple grid search, despite the fact that it is true that PI is not usually sensitive to tuning (Probst et al., 2019).

## 6.3 Experiments on Synthetic Data

### 6.3.1 Motivation

We first evaluate TERC against the baselines using synthetic data, in order to derive controlled and easily interpretable experimental results. Before discussing the results, we provide guidance on their presentation and interpretation. Please refer to Figure 1, which displays six bar graphs. These graphs are organized in a matrix format, with rows representing two different datasets and columns indicating the feature selection method employed. Within each graph, the colored bars denote the 'feature importance' of each state variable during training, as assessed by our method and the two baselines. The red dashed line in each graph marks the upper bound on the score attained by the null model.

State variables whose 'feature importance' values have at least a 95% probability of exceeding this threshold are deemed informative for the learning process and are therefore included in the set of chosen features. This graphical format is consistently applied throughout our analysis when investigating the contribution of state variables to the learning process.

### 6.3.2 Synthetic Data Generation

In this section, we describe the two synthetic datasets under investigation in detail and the nature of the relationships within them. The first one is characterized by CPMCR between four different variables, which we will refer to as the *Four Redundant Variables dataset*. Whereas, the second dataset is characterized by CPMCR between two subsets of variables, which we refer to as the *Two Redundant Triplets dataset*.

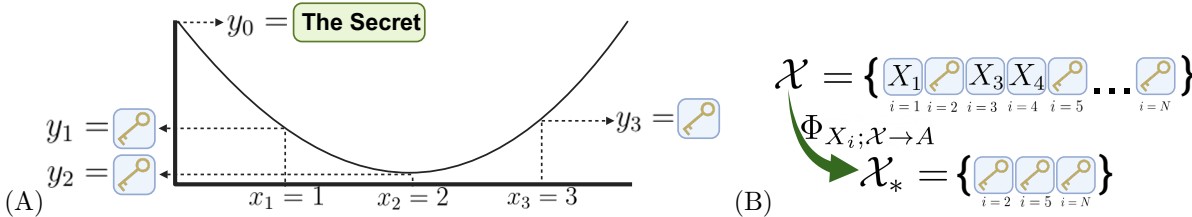

Figure 2: Subfigure (A) illustrates how at least three secret keys are needed to decode a polynomial of order two in Shamir's secure multi-party communication. Subfigure (B) depicts how we use our method of state variable selection to distinguish these three secret-forming keys, from non-secret-forming keys.

The *Four Redundant Variables dataset* is composed of six variables $X_1$, $X_2$, $X_3$, $X_4$, $X_5$, and $X_6$. We generate three stochastic binary arrays of length 10,000 corresponding to the realizations of $X_1, X_2, X_3$. We then derive three arrays for variables $X_4$, $X_5$ and $X_6$ identical to that generated for the variable $X_1$. Consequently, we have perfect redundancy between one of the original binary strings and its three copies. Specifically, we have perfect redundancy between $X_1$, $X_4$, $X_5$, and $X_6$. Finally, we create the target variable $A$, via a three-dimensional XOR function with variables $X_1, X_2$, and $X_3$ as inputs. In other words, a realization of variable $A$ is equal to 1 if the realizations of variables $X_1, X_2$, and $X_3$ are equal, and 0 otherwise. Therefore, our final dataset encompasses not only redundant interactions but also synergistic ones. In order to derive the value of $a^t$, one should only need to know the values of $x_2^t$ and $x_3^t$ and one of either $x_1^t, x_4^t, x_5^t$ or $x_6^t$. Consequently, a method of feature selection should similarly be able to identify variables $X_2$ and $X_3$ and one of either $X_1, X_4, X_5$ or $X_6$. Although this is the case, most current methods for feature selection fail to consider redundancy beyond the pairwise case. Thus, they fail to identify such relationships, as we will demonstrate in Section 6.3.1.

To generate the *Two Redundant Triplets dataset*, we follow an identical scheme. Except here, we let $X_1 \equiv X_4$ while $X_2 \equiv X_5$ and $X_3 \equiv X_6$. We generate the target variable in the same manner as for the previous dataset using a XOR function. This means that now we have CPMCR between the triplet subsets $\mathcal{P} = \{X_1, X_2, X_3\}$ and $\mathcal{P}' = \{X_4, X_5, X_6\}$ (or $\mathcal{P} = \{X_4, X_2, X_3\}$ and $\mathcal{P}' = \{X_1, X_5, X_6\}$ and so on), as $\mathcal{P} \equiv \mathcal{P}'$. A feature selection method should isolate one of these subsets as the minimum number of variables required to calculate $A$.

### 6.3.3   Experimental Results

As depicted in Figure 1, for both synthetic datasets, we see that our method is able to successfully identify the minimal set needed to describe the target variable $A$. This is unlike both UMFI and PI, which fail to identify any variables, because of these methods' inability to resolve the relationships between perfectly redundant variables beyond the pairwise case.

### 6.4   The Secret Key Game

### 6.4.1   Motivation

To clearly demonstrate the application and significance of the newly devised technique within the context of RL, we design a new environment, namely the Secret Key Game. The game is inspired by Shamir's secret-sharing protocol developed for secure multiparty communication. This protocol involves dividing a numerical secret into parts known as secret keys and distributing them among group members (Shamir, 1979; Blakley, 1979). In this section, we introduce this protocol before adapting it to an RL game. We model the problem as follows: the agent learns to calculate the secret from a state of $N$ possible secret keys. However, only three of these keys are actually used to form the secret. The agent's ability to complete its task is consequently impeded until it learns which keys are informative. Therefore, we can leverage our proposed method to exclude non-secret forming keys from the state, resulting in improved game performance. This improvement is attributed to the reduced dimensionality of the state space, in line with Bellman's principles (Bellman & Kalaba, 1959).

We now discuss the secret-sharing protocol that forms the basis for the game. Imagine there is a numerical secret, denoted as $y_0$, which must be kept secure until all members of a pre-determined group agree to disclose it. To accomplish this, we construct a polynomial function $f(x) = y_0 - ax + bx^2$. We then randomly generate the coefficients $a$ and $b$ in the range $[0, 1]$. By doing so, we obtain a curve with a $y$-intercept equal to our secret. We then identify three points belonging to this polynomial curve, specifically $(x_1, y_1), (x_2, y_2)$, and $(x_3, y_3)$, and define the $y$-values of these points ($y_1, y_2$, and $y_3$) as our secret keys to be shared among the group members. Figure 2.A visualizes this process, showing a second-order polynomial curve with three secret keys defined by the $y$-values of three distinct points, while the secret is represented by the $y$-intercept. The original secret can be derived using polynomial interpolation, but if, and only if, all three secret keys are made available. We adapt this method of multi-party communication into a simple RL game by considering the following question: given you had $N$ keys, of which only three were being used to form $y_0$, how many incorrect guesses would it take for an RL agent to learn how to correctly calculate the secret?

The Secret Key Game is played as follows: for each iteration of the game, there is a unique new secret, which is divided into three secret keys. These keys are then hidden among $N - K$ decoy keys to form set $\mathcal{X}$, as illustrated in Figure 2.B. In our experiments, $K = 3$. The set $\mathcal{X}$ then serves as the state of the RL agent, considering the following reward function: $r = -|a^t - y_0|$, i.e., the negative absolute difference between the agent's action at time $t$ and the secret. We design the game as described to showcase the effectiveness of the state variable selection method in detecting non-secret forming keys (i.e., keys that are not inputs of the functions used to generate the secret) in $\mathcal{X}$, resulting in their removal to form $\mathcal{X}_*$ as illustrated in Figure 2.B. This enhances the accuracy of the RL agent in approximating the $y$-intercept of the polynomial function, resulting in improved game performance.

In order to provide a more practical intuition of the game, let us consider a fully worked-out example. To play the secret key game we begin by randomly assigning the secret-forming keys and the decoy keys. Let us suppose the 2nd, 6th, and 25th elements of the state have been randomly assigned as the secret forming values. Furthermore, we assume that we are playing with a state length of 25, pertaining to three secret keys, and 22 decoy keys. In such a scenario, one iteration of the Secret Key Game would proceed as follows: initially, we create a state populated with 25 random integers, each ranging from zero to 10. An example of such a state might be $s^t = [v_{rand_1}, 3, v_{rand_2}, v_{rand_3}, v_{rand_4}, 8...7] \in \mathcal{S}$. We then apply polynomial interpolation to construct a second-order function that intersects the points corresponding to the secret keys $((x = 1, y = 3), (x = 2, y = 8)$, and $(x = 3, y = 7))$. The $y$-values of these points correspond to the random values from the state, and the $x$-values are assigned as either one, two, or three. This process generates a $y$-intercept ($y_0$), equal to -7 in this instance. After that, we reward the RL agent according to a formula we previously introduced in the 'motivation' section, more precisely we have R : $\mathcal{S} \times \mathcal{A} \mapsto [-80..0]$, where $r^t = R(a^t, s^t) = -|a^t - y_0|$, and the range of the potential rewards follows trivially from $y_0 \in \mathcal{A} = [-40..40]$. To proceed with the game, we repeat this process while randomly updating the integers within the state, corresponding to a transition function that returns equal probabilities across all possible states. More formally, we have T : $\mathcal{S} \mapsto \Delta(\mathcal{S})$, where $\Delta$ implies stochastic rather than deterministic transitions. Through repeated exposure, the agent learns a policy $\pi_{\text{sto}} : \mathcal{S} \times \mathcal{A} \to [0, 1]$, such that it maximizes the cumulative reward. It is important to note that the specific indices of the secret keys within the state are not critical, provided they remain consistent throughout the game.

### 6.4.2 Experimental Settings

In our experiments, we consider states formed of 25 or 50 keys when generating trajectories. By assuming that the $y$-values of the secret keys are integers in the range $[0..10]$; the resulting secret is also an integer within the range of $y_0 \in [-40..40]$. Therefore, to ensure that the agent action space is the same as the range of the secret, we employ an RL algorithm with an 80-dimensional discrete action space $\mathcal{A} = [-40..40]$. Due to the large state spaces, we use a one-step temporal difference learning Actor-Critic algorithm (Sutton & Barto, 2018), for steps one and three, as described in 5.4. Finally, as far as the selection of the hyperparameters for the RL algorithm is concerned, we refer to Appendix G.

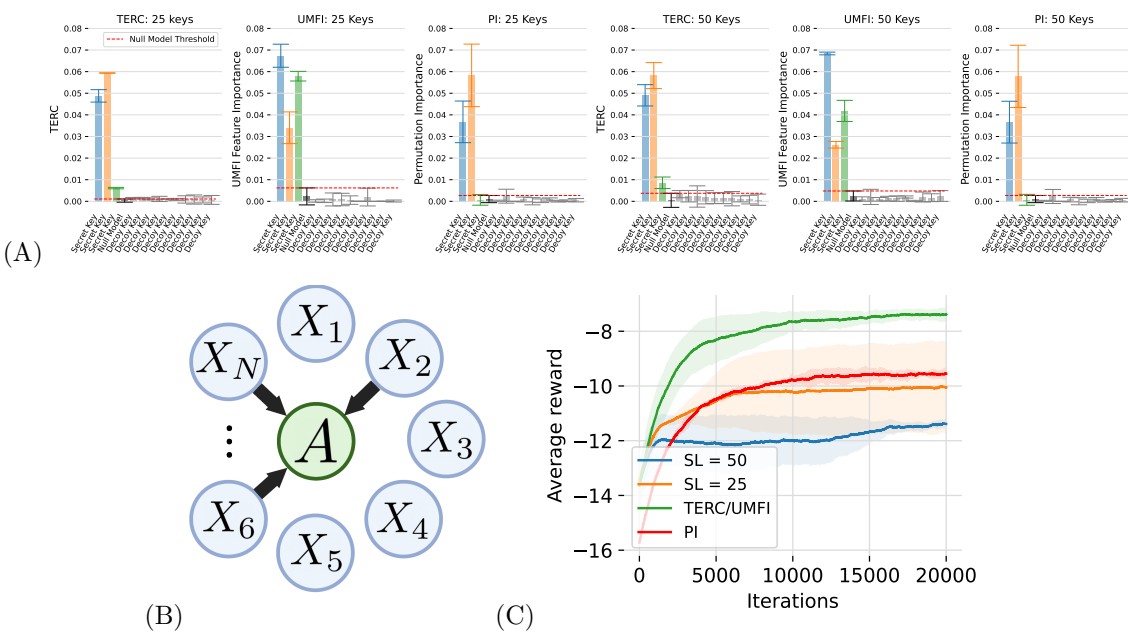

Figure 3: Subfigure (A) depicts the final feature importance values for each key in the secret game when using TERC, UMFI or PI. Subfigure (B) depicts the Bayesian network representation of TERC in the Secret Key Game if the secret keys were at index two, six and $N$. Finally, graph (C) represents how the agent training efficiency varied as a function of state length.

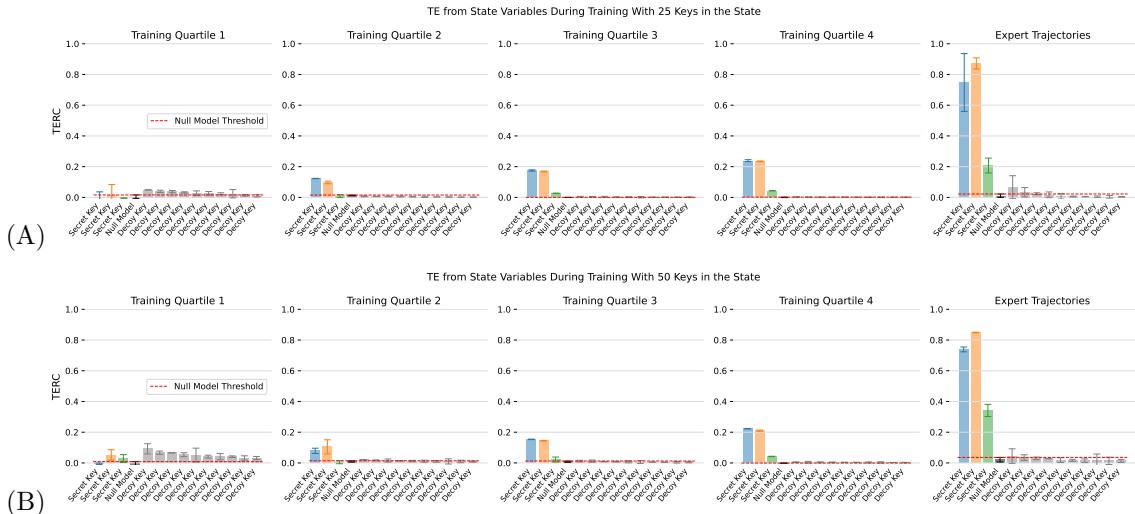

Figure 4: Graphs (A) and (B) show the final values computed when verifying TERC as evaluated during different training quartiles, for 25 and 50 potential secret keys. Similarly to Figure 3, we have included only TERC values for the 10 decoy keys that transferred the most entropy to the actions.

### 6.4.3 Experimental Results

Figure 3.A plots the values of $\Phi_{X_i;\mathcal{X}\rightarrow A}$ that are calculated when aiming to verify TERC. Furthermore, in this plot, we also display values of UMFI and PI for each key. In Figure 3.A, there are 25 potential keys in the state, of which only three form the secret, which we label as secret keys, while the remainder have been labeled as decoy keys. According to the results of TERC in Figure 3.A, the actions were only conditionally dependent on the three secret forming keys, which were randomly selected at the start of the game as keys

$2, 6$ and $25$. We represent the results using a Bayesian network as illustrated in Figure 3.B. UMFI similarly leads to a set consisting of all three secret keys when applied to this simple game. However, PI can only identify two out of three secret keys. We observe similar results in 3.A when the state is composed of 50 keys. Figure 3.C shows how failing to implement state variable selection can deteriorate the agent performance while playing the Secret Key Game. We observe that the state designed using TERC/UMFI achieves the greatest cumulative reward.

Figure 4.A and 4.B depict how $\Phi_{X_i;\mathcal{X} \to A}$ evolves during different stages of training when the state comprises either 25 or 50 keys. As the agent learns, it receives higher rewards by enhancing its ability to decipher the secret. This leads to the agent's actions increasingly depending on the secret keys, as shown in Figures 4.A and 4.B.

### 6.5 Gym Physics Environments

#### 6.5.1 Motivation

In order to show the potential of the proposed method for physics-based RL problems, we use three environments provided by OpenAI Gym (Brockman et al., 2016), namely Cart Pole, Lunar Lander, and Pendulum.

In these settings, the variables that are used to define the state are provided by the environment itself. In order to study the ability of TERC to identify the optimal set of variables, we consider those provided by the environment plus a set of random variables, similarly to (Grooten et al., 2023). For example, in the Cart Pole game[6], the state at a given time $s^t = [x^t, \dot{x}^t, \theta_{pole}^t, \dot{\theta}_{pole}^t]$ is 'doped' with three random variables $v_{rand_i}^t$ to form $s^t = [x^t, \dot{x}^t, \theta_{pole}^t, \dot{\theta}_{pole}^t, v_{rand_1}^t, v_{rand_2}^t, v_{rand_3}^t]$, where $x$, $\theta_{pole}$ , and $\dot{()}$ indicate the $x$-axis position, the pole angle, and the derivative with respect to time, respectively. The state is defined by the set of variables $\mathcal{X} = \{X, \dot{X}, \Theta_{pole}, \dot{\Theta}_{pole}, V_{rand_1}, V_{rand_2}, V_{rand_3}\}$. Subsequently, we train RL agents using this state until convergence and then use the trajectories generated during our analysis (for more details, please refer to Section 3.1). As before, we will demonstrate that, through the methods presented, we can identify the random variables ($V_{rand_i}$) and eliminate them from the state, forming the smallest subset $\mathcal{X}_*$ that preserves information regarding actions. We then demonstrate that by training the RL agents with the set $\mathcal{X}_*$, as described in step three in Section 5.4, we are able to improve the learning efficiency.

Moreover, our analysis reveals that by examining the transfer of entropy at different stages of training, one can infer unique phases in the behavioral dynamics of the system being studied. This implies that TERC could play a significant role in enhancing the interpretability of these systems.

#### 6.5.2 Experimental Settings

For the environments with discrete action and state spaces, namely, Cart Pole and Lunar Lander, we train the agents using a one-step temporal difference Actor-Critic architecture (Sutton & Barto, 2018). Instead, for the continuous action space environment we take into consideration, Pendulum, we use PPO (Schulman et al., 2017). For more details on these methods, including hyperparameters, please refer to Appendix G.

#### 6.5.3 Experimental results

The $\Phi_{X_i;\mathcal{X} \to A}$ and UMFI values of the completely random variables ($V_{rand_i}$), as depicted in Figure 5, fall within the range of the null model. As a result, these random variables are excluded from the set $\mathcal{X}$, to form $\mathcal{X}_*$. Consequently, this updated set contains only the original variables. PI instead fails to detect all the informative variables. Training the agent using the sets identified using TERC, UMFI and PI leads to the learning curves seen on the right-hand side of Figure 5. These graphs show how training can be sped up when using the optimal set of variables as identified using TERC and UMFI.

Now, we examine how $\Phi_{X_i;\mathcal{X} \to A}$ changes as the agent trains, demonstrating TERC's potential use for model interpretability. Figure 6 illustrates these changes as the agent learns the cart-pole game. Initially, the agent learns to maintain the pole's balance by moving in one direction according to the sign of the angular velocity

---

[6]https://gymnasium.farama.org/environments/classic_control/cart_pole/.

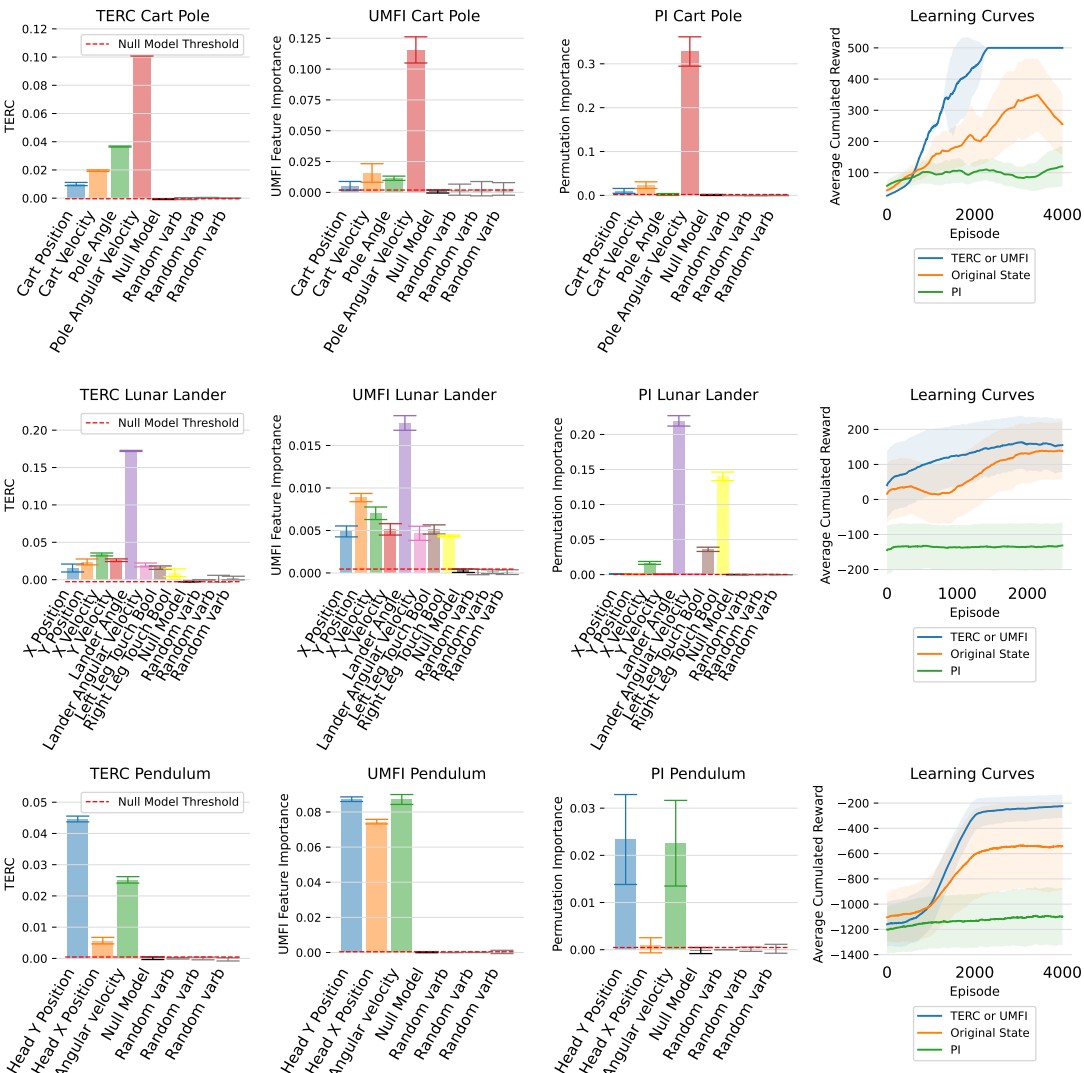

Figure 5: This subfigure depicts the final values obtained for $\Phi_{X_i; \mathcal{X} \to A}$, UMFI, and PI for Cart Pole, Lunar Lander, and Pendulum. On the right-hand side, we illustrate how failing to remove these random variables from the state of the Cart Pole playing agent degrades the performance of the game.

until it reaches the boundary. As a result, during the first quarter of the training, we observe that the agent's actions depend only on the angular velocity. It is possible to observe that the agent tries to avoid the environment's edge during the second quartile of training, which leads to an increase of the value of $\Phi_{X_i; \mathcal{X} \to A}$ for the variables associated with cart velocity and $x$-position. As the agent moves into the fourth quartile, it is increasingly able to remain within the central area of the environment, reducing the need for correcting the cart position; as a result, its actions exhibit less dependence on the cart position variable. In the final graph of Figure 6 (right), the agent has perfected its ability to stay in the center of the environment, making only minor movements to re-adjust the angle of the pole if it deviates from the upright position. This shift in strategy leads to actions that show less dependence on the angular velocity, velocity, and position while exhibiting a stronger dependence on the angle.

Figure 7 instead shows how an RL agent's strategy evolves while learning to play Lunar Lander. Initially, during the first quarter of training, the agent discovers the necessity of taking actions that are dependent on the lander's angle to maintain its upright position. After achieving this, the agent begins to learn to fly around the environment, leading to its actions becoming more dependent upon the speed and position

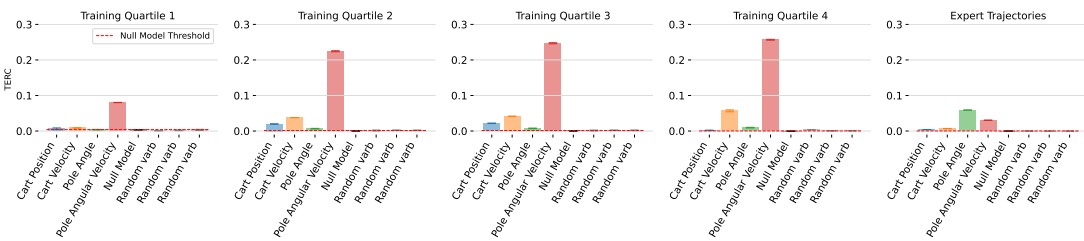

Figure 6: Entropy transferred from the state variables to the actions during different phases of training in the Cart Pole environment.

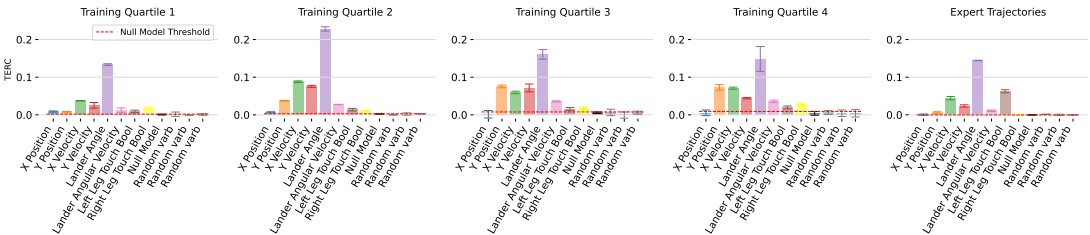

Figure 7: Entropy transferred from the state variables to the actions during different phases of training in the Lunar Lander environment.

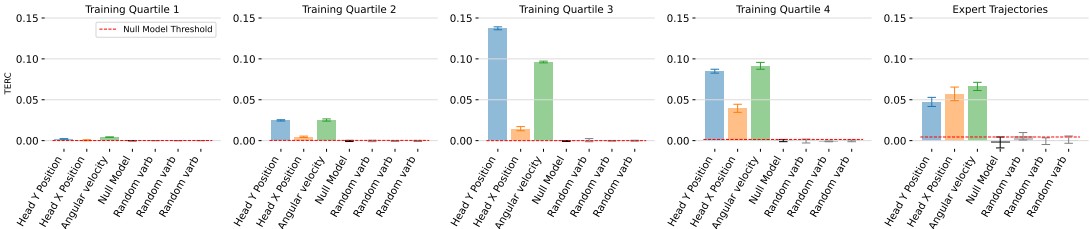

Figure 8: Entropy transferred from the state variables to the actions during different phases of training in the Pendulum environment.

variables throughout the second and third quarters of training. Finally, throughout the trajectories collected from the last quarter of training and the expert trajectories, the agent masters landing within the target area, which results in the actions depending more on the Boolean variable that describes whether the lander's legs have touched down successfully.

Figure 8 demonstrates the changes in $\Phi_{X_i;\mathcal{X}\to A}$ estimates as the agent learns how to play the Pendulum game. During the first quartile, the agent's actions are primarily stochastic due to the lack of a learned strategy, resulting in negligible $\Phi_{X_i;\mathcal{X}\to A}$ values across all variables. In the second and third quartiles, the agent learns to swing the pendulum to the upright position, making its actions increasingly dependent on the $y$-position of the pendulum head and the angular velocity, as depicted in Figures 8.B and 8.C. Once the agent has learned to swing the pendulum upright, it then learns to keep it there by making minor side-to-side adjustments. This is reflected in the values of $\Phi_{X_i;\mathcal{X}\to A}$ reported in the third and fourth quartiles, where we observe a rising dependency on the $x$ position value of the pendulum head.

## 6.6 Tit-For-N-Tats Strategy in the Iterated Prisoner's Dilemma

### 6.6.1 Motivation

The final environment we consider concerns the Iterated Prisoner's Dilemma, a classic matrix game (Axelrod & Hamilton, 1981). Famously, the structure of the rewards in this game leads to tension between adhering

Figure 9: In Subfigure (A) we depict the moves made by both players when playing optimally against TF3T and TF4T strategies in the Iterated Prisoner's Dilemma. The colored boxes indicate the minimum history needed to learn this optimal strategy, while the colored arrows indicate one period of a repeating action sequence, where $C$ stands for cooperative moves, while $D$ represents defective ones. In Subfigure (B), we illustrate schematically how our measure is used to determine the optimal state history length of three, out of a maximum of nine, when facing a TF4T opponent.

to either competitive or cooperative strategies. Players choose to cooperate with or defect against their opponent, according to the reward matrix shown in Appendix G.6.

We demonstrate our method can be used to discover optimal history lengths when temporally extended states are required. To do this, we first discuss how the Iterated Prisoner's Dilemma is represented as an RL problem. The state ($s^t \in \mathcal{S}$) is commonly formed of the past $l$ actions ($a_i^t \in \{C, D\} = \mathcal{A}$) of the player and their opponent (Anastassacos et al., 2020). For example, let us assume $l = 2$ and Player 1 had defected the last two turns, whereas Player 2 had cooperated: in this case, the state would be defined as $s^t = [D, C, D, C] \in \mathcal{S}$, where $C$ and $D$ represent cooperative and defective moves respectively. Subsequently, a player updates its action based on its current state ($\pi(a^t|s^t) : \mathcal{S} \mapsto \mathcal{A}$) in a manner that maximizes the total cumulative reward. Meanwhile, the reward received and the state transition function depend only on the actions, more precisely R $: \mathcal{A}_1 \times \mathcal{A}_2 \mapsto \{0, 1, 2, 3\}$ and T $: \mathcal{A}_1 \times \mathcal{A}_2 \mapsto \mathcal{S}$. It has been shown that if all contestants are attempting to maximize their reward according to the matrix shown in Appendix G.6, the most recent action of the agent and their opponent, or a state of history length $l = 1$, is sufficient to learn optimal strategies (Press & Dyson, 2012). On the contrary, if the opponent is playing according to other strategies, this is not the case. One such example is when the opponent plays a TFNT (Tit-For-N-Tats) strategy; here, we can use our approach to select the length of the state. This situation is representative of a class of problems in which the history length plays a key role in the definition of the state.

We assume that the opponent plays according to a TFNT (Tit-For-N-Tats) strategy. This is analogous to the Tit-For-Tat strategy, in which the player mirrors their opponent's last move. However, in this case, TFNT only defects in response to $N$ consecutive defective actions from their opponent, while otherwise cooperating. Against such strategies, the optimal way of playing is taking $N - 1$ defective actions in a row before then cooperating. To play this optimal strategy agents must consequently have a minimum history length of $N - 1$. This is illustrated by means of the red boxes in Figure 9.A, in which we show the optimal opponent strategies (OOS) against Tit-For-3-Tats (TF3T) and Tit-For-4-Tats (TF4T) strategies. For example, against TF3T the OOS only cooperates after they have defected for the last two iterations while the opponent cooperated. In this final experiment, we will show that we can reliably identify the optimal state history length using TERC.

### 6.6.2 Experimental Settings

As previously stated, we let $l$ denote the maximum history length of interest. We then generate training trajectories by having a tabular Q-learning agent (with hyperparameters available in Appendix G) use a state with history length $l$ to learn to play against a TFNT opponent. From the resulting trajectories, we form the set $\mathcal{X} = \{X_1...X_l\}$, as shown in Figure 9.A (see Section 3.1). The variables ($X_i$) in Figure 9.A represent action pairs ordered in time. Additionally, we choose $L = 9$ because we assume that, in this example, we are interested in investigating strategies up to Tit-For-10-Tats.

If the opponent is playing TFNT, where $N < 5$, the optimal strategy of defecting $N - 1$ times before cooperating, is repeated in a cyclic fashion within $\mathcal{X}$. For example, if playing optimally against a Tit-For-3-Tats (TF3T) opponent, the realizations of variable $X_i$ and $X_{i+3}$ are the same. This is because the optimal strategy against such an opponent is to cooperate every third move, whilst otherwise defecting. Consequently,

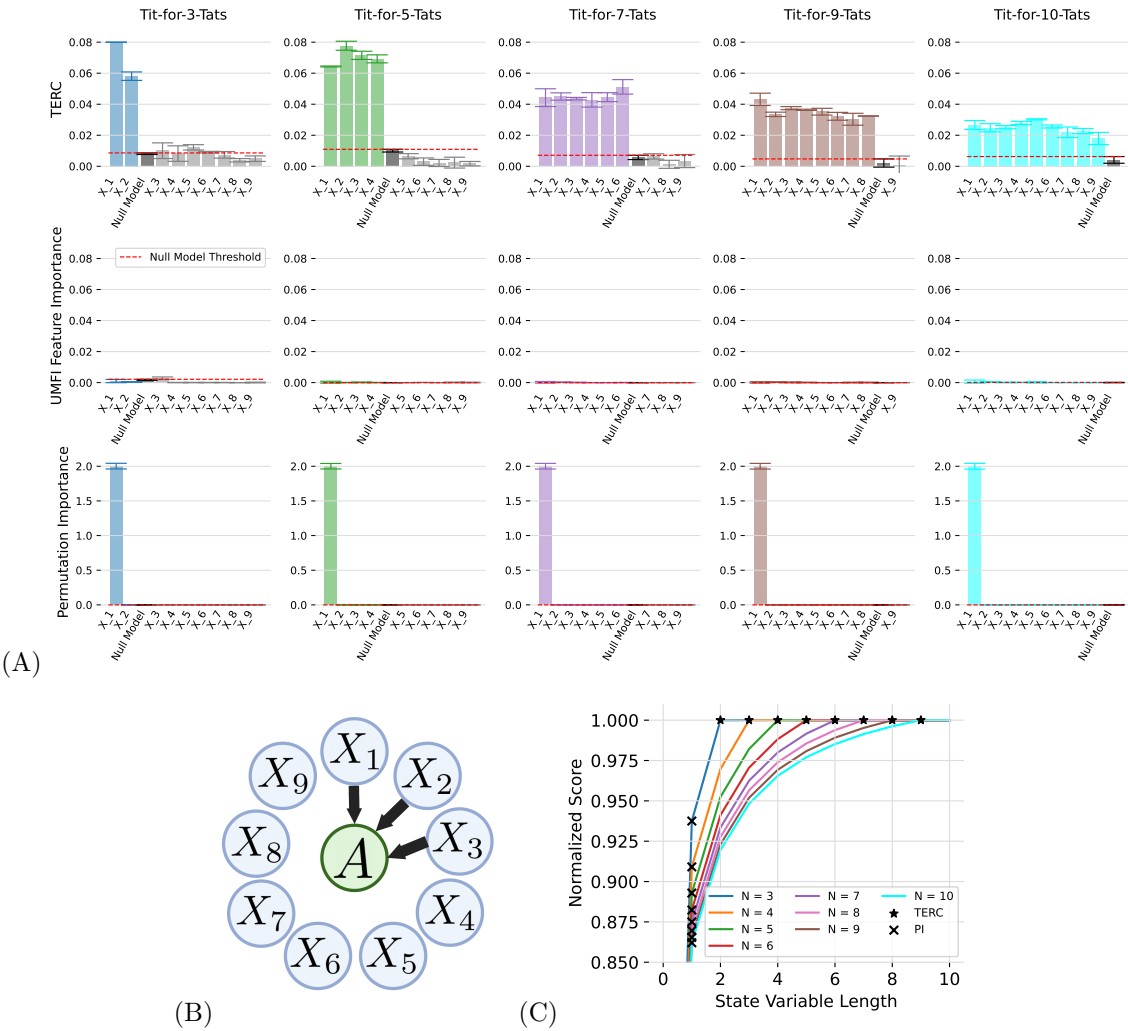

(A)

(B)                                        (C)

Figure 10: In Subfigure (A) we compare $\Phi_{X_i;\mathcal{X}\rightarrow A}$, UMFI and PI, values for all state variables from $l = 1$ to $l = 9$, when playing optimally against a TFNT opponent. For clarity of presentation, we only present the results with the values of $N$ displayed above. These results were representative of how our algorithm performed for all TFNT opponents. Subfigure (B) shows the resulting Bayesian network representation of how the actions depend on state variables against a TF4T opponent. Subfigure (C) shows how the change of the state size affects an agent's performance against a TFNT strategy. We use stars to denote the optimal state length highlighted by our method, whereas we use crosses to represent the state length generated using PI. Finally, we omit the results for UMFI in Subfigure (C) for clarity.

this strategy is cyclic with a period of length 3. This is represented in the form of colored arrows in Figure 9.A. Although we have presented our argument when playing against a TF3T opponent, this strategic behavior does generalize. In other words, when playing optimally against a TFNT strategy, the realizations of variables $X_i$ and $X_{i+N}$ are the same, and their information-theoretic properties are identical. Therefore, we have CPMCR such that $\Psi_{X_i,X_{i+N}}(A|\mathcal{X})$. Furthermore, we not only observe CPMCR between $X_i$ and $X_{i+N}$, but also between variables $X_i$, $X_{i+N}$, and $X_{i+2N}$, and between subsets $\mathcal{P} = \{X_i, X_{i+1} \dots X_{i+N-1}\}$, and $\mathcal{P} = \{X_{i+N}, X_{i+N+1} \dots X_{i+2N-1}\}$. Essentially, these variables and their associated datasets are characterized by complex redundancies. With respect to the selection of the hyperparameters and network structure used during Algorithm 1, we refer to Appendix G.

### 6.6.3 Experimental Results

In Figure 10.A, we show the values of $\Phi_{X_i;\mathcal{X}\to A}$, UMFI and PI when applied to the trajectories generated in the case of a Q-learner play against a TFNT opponent. In these datasets we do not have simple pairwise redundancy and, for this reason, the assumptions in (Janssen et al., 2023) are not verified. Therefore, this method is not able to identify the correct set of variables; PI is similarly ineffective. Only TERC has the ability to identify the correct set of variables. We represent the values in the TF4T case pictorially as the Bayesian network shown in Figure 10.A. This is further reflected in the agent performance results reported in Figure 10.C, where we plot the normalized cumulative reward achieved over the last 1000 iterations of the game, as a function of the length of the state history. In Figure 10.C we report the percentage of times the agent plays optimally during the last 1000 iterations of training. Agents trained with a history length smaller than $N-1$ converge to suboptimal strategies. Consequently, we observe that only state lengths equal to 9 achieve optimal cumulative rewards against the TF10T opponent, whereas state lengths of 2 play optimally against the TF3T opponent. The variables that satisfy TERC in Figure 10.C can therefore be interpreted as indicating the minimum subset of $\mathcal{X}$ required to play optimally, corroborating the theoretical observations reported earlier. The detailed learning dynamics are reported in the Appendix F.

## 7 Conclusions

In this article, we have introduced a novel information-theoretic methodology for state variable selection in RL, based on TERC, a criterion that can be used to verify the existence of dependencies between state variables and the agent's actions. The objective is the definition of the minimal subset of state variables from which an agent could learn optimal policies. We have defined this optimal subset as one whose realizations reduce the entropy of the actions identically to those of the set of all potential state variables.

We have demonstrated that, assuming CPMCR is not verified, a naïve application of TERC is sufficient to identify the minimal set (Theorem 1). However, this assumption is not universally satisfied. Consequently, we have also presented a solution in case of CPMCR. We have presented an implementation, underpinned by theoretical results, which guarantees the derivation of the minimal set with a weak assumption (Theorem 2). In general, TERC represents a novel approach toward feature selection that scales linearly in time without relying on the restricting assumption of having only pairwise dependencies in the set of variables. Furthermore, even if $C_1$ is not satisfied, we still select an information-theoretically optimal set; but we cannot ensure it is also minimal.

Finally, we have implemented our method using a neural estimator and we have then evaluated TERC using synthetic data and across diverse environments representative of various RL problem classes. In particular, we have demonstrated that our proposed method consistently identifies the optimal set of variables in presence of redundant and synergestic relationships.

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

## A  Algorithm for the Estimation of Transfer Entropy Based Measure

In this section, we outline an algorithm for the estimation of the transfer entropy based measure defined in Equation 6.

---

**Algorithm 3** Estimation of $\Phi_{X_i;\mathcal{X}\to A}$.

---

**Input**: Training trajectories $\tau = (s^1, a^1, s^2, a^2 ... s^T, a^T) = \{\mathcal{X}, A\}$, where $a^1, a^2 \in A$ and $s^1, s^2 \in \mathcal{X}$. Target variable $X_i$.

**Output**: $H(A|\mathcal{X}_{\setminus X_i}) - H(A|\mathcal{X})$

1: Initialize weights for $\theta$ and $\theta_{\setminus X_i}$
2: **for** $1$ to $N$ **do**
3:   Draw mini batch samples of length $b$ from the joint distribution of the actions and the state with all possible variables included $p_{A,\mathcal{X}} \sim (a^{t_1}, x_1^{t_1}, x_i^{t_1} \dots x_N^{t_1}), \dots, (a^{t_b}, x_1^{t_b}, x_i^{t_b} \dots x_N^{t_b})$, and repeat for the marginal distribution $p_A \otimes p_\mathcal{X} \sim (a^{t'_1}, x_1^{t_1}, x_i^{t_1} \dots x_N^{t_1}), \dots, (a^{t'_b}, x_1^{t_b}, x_i^{t_b} \dots x_N^{t_b})$, where $t'_i \neq t_i$.
4:   Draw mini batch samples of length $b$ from the joint distribution of the actions and the state with variable $X_i$ missing. $p_{A,\mathcal{X}_{\setminus X_i}} \sim (a^{t_1}, x_1^{t_1} \dots x_N^{t_1}), \dots, (a^{t_b}, x_1^{t_b} \dots x_N^{t_b})$, and repeat for the marginal distribution $p_A \otimes p_{\mathcal{X}_{\setminus X_i}} \sim (a^{t'_1}, x_1^{t_1}, \dots x_N^{t_1}), \dots, (a^{t'_b}, x_1^{t_b} \dots x_N^{t_b})$, where $t'_i \neq t_i$.
5:   $I(A; \mathcal{X}) = \frac{1}{b} \sum_{j=1}^b F_\theta(a^{t_j}, x_1^{t_j}, x_i^{t_j} \dots x_N^{t_j})) - \frac{1}{b} \sum_{j=1}^b \log e^{F_\theta((a_1^{t'_j}, x_1^{t_j}, x_i^{t_j} \dots x_N^{t_j})}$
6:   $I(A; \mathcal{X}_{X_i}) = \frac{1}{b} \sum_{j=1}^b F_{\theta_{\setminus X_i}}((a^{t_j}, x_1^{t_j} \dots x_N^{t_j})) - \frac{1}{b} \sum_{j=1}^b \log e^{F_{\theta_{\setminus X_i}}(a_1^{t'_j}, x_1^{t_j} \dots x_N^{t_j})}$
7: **end for**
8: **return** $I(A; \mathcal{X}) - I(A; \mathcal{X}_{\setminus X_i})$

---

## B  Proof of Non-Negativity of $\Phi_{X_i\mathcal{X}\to A}$

In this section, we prove the non-negativity of $\Phi_{X_i;\mathcal{X}\to A}$. We first write the full expression of the measure as follows:

$$\Phi_{X_i;\mathcal{X}\to A} = -\int_{A \times \mathcal{X}} p_{A,\mathcal{X}}(a^t, x_1^t, x_i^t \dots x_N^t) \log \frac{p_{A,\mathcal{X}_{\setminus X_i}}(a^t, x_1^t \dots x_N^t), p_\mathcal{X}(x_1^t, x_i^t \dots x_N^t)}{p_{A,\mathcal{X}}(a^t, x_1^t, x_i^t \dots x_N^t), p_{\mathcal{X}_{\setminus X_i}}(x_1^t \dots x_N^t)} dA \times \mathcal{X}. \qquad (9)$$

By applying Jensen's inequality we can then write:

$$\begin{aligned}
\Phi_{X_i;\mathcal{X}\to A} &\geq -\log \int_{A \times \mathcal{X}} p_{A,\mathcal{X}}(a^t, x_1^t, x_i^t \dots x_N^t) \frac{p_{A,\mathcal{X}_{\setminus X_i}}(a^t, x_1^t \dots x_N^t), p_\mathcal{X}(x_1^t, x_i^t \dots x_N^t)}{p_{A,\mathcal{X}}(a^t, x_1^t, x_i^t \dots x_N^t), p_{\mathcal{X}_{\setminus X_i}}(x_1^t \dots x_N^t)} dA \times \mathcal{X} \\
&\geq -\log \int_{A \times \mathcal{X}} \frac{p_{A,\mathcal{X}_{\setminus X_i}}(a^t, x_1^t \dots x_N^t), p_\mathcal{X}(x_1^t, x_i^t \dots x_N^t)}{p_{\mathcal{X}_{\setminus X_i}}(x_1^t \dots x_N^t)} dA \times \mathcal{X} \\
&\geq -\log(1) \\
&\geq 0,
\end{aligned} \qquad (10)$$

therefore proving the non-negativity of $\Phi_{X_i;\mathcal{X}\to A}$. $\square$

## C  Proof of Lemma 1

The steps of this proof can be summarized as follows. Firstly, we show that a single element of a subset for which we observe CPMCR with another variable or subset of variables will not be included in the set $\mathcal{X}_\Phi$. Then, we will demonstrate that this applies to all the elements of both subsets. Finally, we use the definition of CPMCR to prove Lemma 1.

Let there be a case of CPMCR between two subsets $\mathcal{P}, \mathcal{P}' \subseteq \mathcal{X}$ such that:

$$
\begin{aligned}
H(A|\mathcal{X}_{\backslash(\mathcal{P},\mathcal{P}')} \cup \mathcal{P} \cup \mathcal{P}') &= H(A|\mathcal{X}_{\backslash(\mathcal{P},\mathcal{P}')} \cup \mathcal{P}) \\
&= H(A|\mathcal{X}_{\backslash(\mathcal{P},\mathcal{P}')} \cup \mathcal{P}') \\
&< H(A|\mathcal{X}_{\backslash(\mathcal{P},\mathcal{P}')}).
\end{aligned}
\tag{11}
$$

Let $\mathcal{P}_{\backslash\mathcal{P}} \in \mathscr{P}(\mathcal{X}_{\backslash\mathcal{P}})$ and $\mathcal{P}_{\backslash\mathcal{P}'} \in \mathscr{P}(\mathcal{X}_{\backslash\mathcal{P}'})$. Given Equation 2 and Equation 11, it must be true that:

$$
H(A|\mathcal{P}_{\backslash\mathcal{P}'} \cup \mathcal{P}') \leq H(A|\mathcal{P}_{\backslash\mathcal{P}'} \cup \{P'\}),
\tag{12}
$$

where $P' \in \mathcal{P}'$ . Combining Equations 11 and 12 leads us too:

$$
H(A|\mathcal{P}_{\backslash\mathcal{P}} \cup \mathcal{P}) \leq H(A|\mathcal{P}_{\backslash\mathcal{P}} \cup \{P'\}),
\tag{13}
$$

we now substitute in $\mathcal{X}_{\backslash(\mathcal{P},P')} = \mathcal{P}_{\backslash\mathcal{P}}$, leading to:

$$
\begin{aligned}
H(A|\mathcal{X}_{\backslash(\mathcal{P},P')} \cup \mathcal{P}) &\leq H(A|\mathcal{X}_{\backslash(\mathcal{P},P')} \cup \{P'\}) \\
H(A|\mathcal{X}_{\backslash P'}) &\leq H(A|\mathcal{X}_{\backslash\mathcal{P}}) \\
&\leq H(A|\mathcal{X}),
\end{aligned}
\tag{14}
$$

and therefore:

$$
H(A|\mathcal{X}_{\backslash P'}) - H(A|\mathcal{X}) \leq 0.
\tag{15}
$$

Due to the non-negativity proof presented in Section B, we obtain $\Phi_{P';\mathcal{X}\to A} = 0$, and consequently element $P'$ is not included in the set $\mathcal{X}_\Phi$. This holds $\forall P \in \mathcal{P}$ and $\forall P' \in \mathcal{P}'$. Consequently, neither of the elements of the subset $\mathcal{P}$ or of the subset $\mathcal{P}'$ will be included in the set $\mathcal{X}_\Phi$. Therefore, $\mathcal{X}_\Phi = \mathcal{X}_{\backslash(\mathcal{P},\mathcal{P}')}$, but, through Equation 11, we obtain $H(A|\mathcal{X}_\Phi) > H(A|\mathcal{X})$. □

## D   Proof of Theorem 1

We outline this proof in three steps, which we present as lemmas. The first of these lemmas characterizes the variables that are not to be included in $\mathcal{X}_\Phi$. By means of the other two, we demonstrate that by not adding these variables to the set $\mathcal{X}_\Phi$, we still satisfy $H(A|\mathcal{X}) = H(A|\mathcal{X}_\Phi)$.

**Lemma 2.** *Let us assume that there exists variables $X_i$ in $\mathcal{X}$ that transfers no entropy to the actions $A$ (satisfying $\Phi_{X_i;\mathcal{X}\to A} = 0$).*

- *These variables satisfy one of either:*

$$
H(A|\mathcal{X}) = H(A|\mathcal{X}_{\backslash X_i}) < H(A|X_i) < H(A)),
\tag{16}
$$

   *or*

$$
H(A|\{X_i\} \cup \mathcal{P}_{\backslash X_i}) = H(A|\mathcal{P}_{\backslash X_i}));
\tag{17}
$$

- *assuming no cases of CPMCR ($\neg\Psi(A|\mathcal{X})$), the following also holds:*

$$
\begin{aligned}
\Phi_{X_i;\mathcal{X}\to A} = 0 \quad \& \quad \neg\Psi(A|\mathcal{X}) \Leftrightarrow & H(A|\mathcal{X}) = H(A|\mathcal{X}_{\backslash X_i}) < H(A|X_i) < H(A) \quad \text{or} \\
& H(A|\{X_i\} \cup \mathcal{P}_{\backslash X_i}) = H(A|\mathcal{P}_{\backslash X_i}).
\end{aligned}
\tag{18}
$$

*Proof.* Let us assume a case of CPMCR such that $\Psi_{X_i,\mathcal{P}'}(A|\mathcal{X})$, as this naturally leads to $\Phi_{X_i;\mathcal{X}\to A} = 0$. We now show that to satisfy the condition $\neg\Psi_{X_i,\mathcal{P}'}(A|\mathcal{X})$, the variable $X_i$ must satisfy Equation 16 or Equation 17.

We first note that we can re-write $\Phi_{X_i;\mathcal{X}\to A} = 0$ as $H(A|\mathcal{X}_{\setminus X_i}) = H(A|\mathcal{X})$ and therefore the condition $\Phi_{X_i;\mathcal{X}\to A} = 0$ is already satisfied in the definition of CPMCR. Consequently, it is possible to write:

$$
\begin{aligned}
&\Phi_{X_i;\mathcal{X}\to A} = 0 \quad \& \quad \Psi_{X_i,\mathcal{P}'}(A|\mathcal{X}) \\
\Leftrightarrow & (X_i \in \mathcal{X}, \exists \mathcal{P}' \in \mathscr{P}(\mathcal{X}_{\setminus X_i}) : H(A|\mathcal{X}) = H(A|\mathcal{X}_{\setminus \mathcal{P}'}) = H(A|\mathcal{X}_{\setminus X_i}) < H(A|\mathcal{X}_{\setminus (X_i,\mathcal{P}')}) \quad \& \\
& \psi_{\mathcal{P}'}).
\end{aligned}
\tag{19}
$$

Suppose that we violate the condition $\psi'_{\mathcal{P}}$ by adding a non-informative variable $V_{rand}$ to the set $\mathcal{P}'$. Although, this would lead to $\neg\Psi_{X_i,\mathcal{P}'}(A|\mathcal{X})$, it would still be true that $\Psi_{X_i,\mathcal{P}'_{\setminus V_{rand}}}(A|\mathcal{X})$. Consequently, there will always exist $\mathcal{P}' \in \mathscr{P}$ such that $\psi'_{\mathcal{P}}$ is satisfied. Given there always exists a scenario in which $\psi'_{\mathcal{P}}$ is true, it cannot be used to induce the condition $\neg\Psi_{X_i,\mathcal{P}'}(A|\mathcal{X})$. Consequently, for clarity, we remove it from the presentation and we write:

$$
\begin{aligned}
&\Phi_{X_i;\mathcal{X}\to A} = 0 \quad \& \quad \Psi_{X_i,\mathcal{P}'}(A|\mathcal{X}) \\
\Leftrightarrow & (X_i \in \mathcal{X}, \exists \mathcal{P}' \in \mathscr{P}(\mathcal{X}_{\setminus X_i}) : H(A|\mathcal{X}) = H(A|\mathcal{X}_{\setminus \mathcal{P}'}) = H(A|\mathcal{X}_{\setminus X_i}) < H(A|\mathcal{X}_{\setminus (X_i,\mathcal{P}')})).
\end{aligned}
\tag{20}
$$

The next step in this proof is to rewrite Equation 20 so that it satisfies $\neg\Psi_{X_i,\mathcal{P}'}(A|\mathcal{X})$. This will be the case if the equality that relates $H(A|\mathcal{X}_{\setminus \mathcal{P}'}) = H(A|\mathcal{X}_{\setminus X_i})$ or $H(A|\mathcal{X}_{\setminus X_i}) < H(A|\mathcal{X}_{\setminus (X_i,\mathcal{P}')})$ no longer holds. These are the only ways to derive $\neg\Psi_{X_i,\mathcal{P}'}(A|\mathcal{X})$ without leading to $\neg\Phi_{X_i;\mathcal{X}\to A}$.

We now assume the equality that relates $H(A|\mathcal{X}_{\setminus \mathcal{P}'}) = H(A|\mathcal{X}_{\setminus X_i})$ no longer holds. We obtain a variable $X_i$ and subset $\mathcal{P}'$ that do not provide the same information about the target variable $H(A|\mathcal{X}_{\setminus \mathcal{P}'}) > H(A|\mathcal{X}_{\setminus X_i})$. Substituting this into Equation 20 we have:

$$
\begin{aligned}
&\Phi_{X_i;\mathcal{X}\to A} = 0 \quad \& \quad \neg\Psi_{X_i,\mathcal{P}'}(A|\mathcal{X}) \\
\Leftarrow & (X_i \in \mathcal{X}, \exists \mathcal{P}' \in \mathscr{P}(\mathcal{X}_{\setminus X_i}) : H(A|\mathcal{X}) = H(A|\mathcal{X}_{\setminus X_i}) < H(A|\mathcal{X}_{\setminus \mathcal{P}'}) < H(A|\mathcal{X}_{\setminus (X_i,\mathcal{P}')})),
\end{aligned}
\tag{21}
$$

Suppose we rewrite Equation 21 letting $\mathcal{P}' = \mathcal{X}_{\setminus X_i}$. We obtain:

$$
\begin{aligned}
&\Phi_{X_i;\mathcal{X}\to A} = 0 \quad \& \quad \neg\Psi_{X_i,\mathcal{P}'}(A|\mathcal{X}) \\
\Leftarrow & (H(A|\mathcal{X}) = H(A|\mathcal{X}_{\setminus X_i}) < H(A|\mathcal{X}_{\setminus \mathcal{X}_{\setminus X_i}}) \leq H(A|\mathcal{X}_{\setminus \{X_i,\mathcal{X}_{\setminus X_i}\}})) \\
\Leftarrow & (H(A|\mathcal{X}) = H(A|\mathcal{X}_{\setminus X_i}) < H(A|X_i) < H(A)).
\end{aligned}
\tag{22}
$$

Hence, the variables that satisfy Equation 17 also satisfy $\Phi_{X_i;\mathcal{X}\to A} = 0 \quad \& \quad \neg\Psi_{X_i,\mathcal{P}'}(A|\mathcal{X})$. Now, let us suppose that the inequality $H(A|\mathcal{X}_{\setminus X_i}) < H(A|\mathcal{X}_{\setminus (X_i,\mathcal{P}')})$ no longer holds. Specifically, we assume neither the subset $\mathcal{P}'$ or variable $X_i$ provide any information about the target variable. In this case it is true that $H(A|\mathcal{X}_{\setminus X_i}) = H(A|\mathcal{X}_{\setminus (X_i,\mathcal{P}')})$ and we can write the following:

$$
\begin{aligned}
&\Phi_{X_i;\mathcal{X}\to A} = 0 \quad \& \quad \neg\Psi_{X_i,\mathcal{P}'}(A|\mathcal{X}) \\
\Leftarrow & (X_i \in \mathcal{X}, \exists \mathcal{P}' \in \mathscr{P}(\mathcal{X}_{\setminus X_i}) : H(A|\mathcal{X}) = H(A|\mathcal{X}_{\setminus X_i}) = H(A|\mathcal{X}_{\setminus \mathcal{P}'}) = H(A|\mathcal{X}_{\setminus (X_i,\mathcal{P}')})).
\end{aligned}
\tag{23}
$$

We now substitute in $\mathcal{P}_{\setminus (X_i,\mathcal{P}')} = \mathcal{X}_{\setminus (X_i,\mathcal{P}')}$ in Equation 23 such that:

$$
\begin{aligned}
&\Phi_{X_i;\mathcal{X}\to A} = 0 \quad \& \quad \neg\Psi_{X_i,\mathcal{P}'}(A|\mathcal{X}) \\
\Leftarrow & (X_i \in \mathcal{X}, \exists \mathcal{P}' \in \mathscr{P}(\mathcal{X}_{\setminus X_i}) : H(A|\mathcal{X}) = H(A|\mathcal{P}_{\setminus (X_i,\mathcal{P}')} \cup \mathcal{P}') = H(A|\mathcal{P}_{\setminus (X_i,\mathcal{P}')} \cup X_i) = H(A|\mathcal{P}_{\setminus (X_i,\mathcal{P}')}).
\end{aligned}
\tag{24}
$$

Given $\mathcal{P}' \in \mathscr{P}(\mathcal{X}_{\setminus X_i})$ and $\mathcal{P}_{\setminus (X_i,\mathcal{P}')} \in \mathscr{P}(\mathcal{X}_{\setminus (X_i,\mathcal{P}')})$, it must be so that $\mathcal{P}_{\setminus (X_i,\mathcal{P}')} \cup \mathcal{P}' \in \mathscr{P}(\mathcal{X}_{\setminus X_i})$. Consequently, we replace $\mathcal{P}_{\setminus (X_i,\mathcal{P}')} \cup \mathcal{P}'$ with $\mathcal{P}_{\setminus X_i}$ in Equation 24 to give

$$
\begin{aligned}
&\Phi_{X_i;\mathcal{X}\to A} = 0 \quad \& \quad \neg\Psi_{X_i,\mathcal{P}'}(A|\mathcal{X}) \Leftarrow (X_i \in \mathcal{X}, \exists \mathcal{P}' \in \mathscr{P}(\mathcal{X}_{\setminus X_i}) : \\
& \qquad H(A|\mathcal{P}_{\setminus X_i} \cup X_i) = H(A|\mathcal{P}_{\setminus X_i}) = H(A|\mathcal{P}_{\setminus (X_i,\mathcal{P}')} \cup X_i) = H(A|\mathcal{P}_{\setminus (X_i,\mathcal{P}')})).
\end{aligned}
\tag{25}
$$

Since $\mathcal{P}_{\setminus (X_i,\mathcal{P}')} \subseteq \mathcal{P}_{\setminus X_i}$, we do not include the last two equivalences as they are sub-conditions of the first equivalence. Therefore, we can write

$$
\Phi_{X_i;\mathcal{X}\to A} = 0 \quad \& \quad \neg\Psi_{X_i,\mathcal{P}'}(A|\mathcal{X}) \Leftarrow H(A|\mathcal{P}_{\setminus X_i} \cup X_i) = H(A|\mathcal{P}_{\setminus X_i})).
\tag{26}
$$

Combining the statements in Equations 26 and 22 completes the proof of Lemma 2. $\square$

According to Lemma 2, variables that satisfy Equation 17 are not included in the set $\mathcal{X}_\Phi$, despite this, we now prove that $H(A|\mathcal{X}) = H(A|\mathcal{X}_\Phi)$ still holds.

**Lemma 3.** *Let us assume there exists a non-empty subset of variables in $\mathcal{X}$ that satisfies Equation 16. The following holds: $H(A|\mathcal{X}) = H(A|\mathcal{X}_\Phi)$.*

*Proof.* To prove Lemma 3, we let variables $X_j$ and $X_k$ satisfy the property expressed by Equation 17, and show that despite removing them from $\mathcal{X}$ to form $\mathcal{X}_\Phi$, $H(A|\mathcal{X}) = H(A|\mathcal{X}_\Phi)$ is still satisfied. We then demonstrate that this result generalizes to cases with more than two variables.

In accordance with Lemma 2, if we remove $X_j$ from $\mathcal{X}$ we have $H(A|\mathcal{X}) = H(A|\mathcal{X}_{\setminus X_j})$. Suppose that we now repeat this process, but instead, we remove $X_k$ from $\mathcal{X}_{\setminus X_j}$, as this will obtain $\mathcal{X}_\Phi$. To begin, we let $\mathcal{P}_{\setminus X_k} \in \mathscr{P}(\mathcal{X}_{\setminus X_k})$, such that:

$$H(A|\mathcal{P}_{\setminus X_k}) = H(A|\{X_k\} \cup \mathcal{P}_{\setminus X_k}). \tag{27}$$

We now substitute it in $\mathcal{P}_{\setminus X_k} = \mathcal{X}_{\setminus \{X_k, X_j\}}$:

$$H(A|\mathcal{X}_{\setminus \{X_k, X_j\}}) = H(A|\{X_k\} \cup \mathcal{X}_{\setminus \{X_k, X_j\}}), \tag{28}$$

because $\{X_k\} \cup \mathcal{X}_{\setminus \{X_k, X_j\}} = \mathcal{X}_{\setminus X_j}$ we have:

$$\begin{aligned} H(A|\mathcal{X}_{\setminus \{X_k, X_j\}}) &= H(A|\mathcal{X}_{\setminus X_j}) \\ &= H(A|\mathcal{X}). \end{aligned} \tag{29}$$

However, given that only the variables $X_j$ and $X_k$ satisfy the property expressed by Equation 17, we have $\mathcal{X}_{\setminus \{X_k, X_j\}} = \mathcal{X}_\Phi$ and therefore $H(A|\mathcal{X}_\Phi) = H(A|\mathcal{X})$. Consequently, we have proved Lemma 3 in the two variable case.

We now generalize this proof beyond the two variable case. Let variable $X_l$ also satisfy the property outlined in Equation 17, we now have $\mathcal{X}_{\setminus \{X_k, X_j, X_l\}} = \mathcal{X}_\Phi$. We complete this proof, by first using the property outlined in Equation 17 to show $H(A|\mathcal{X}_{\setminus \{X_k, X_j\}}) = H(A|\mathcal{X}_{\setminus \{X_k, X_j, X_l\}})$, before then combining this with the proof for the two variable case to show $H(A|\mathcal{X}_\Phi) = H(A|\mathcal{X})$. We can repeatedly apply this proof up to the $N$ variable case. $\square$

According to Lemma 2, the variables that satisfy Equation 16 are not included in the set $\mathcal{X}_\Phi$. Despite this, provided no cases of CPMCR exist in the set $\mathcal{X}$, it is possible to prove that $H(A|\mathcal{X}) = H(A|\mathcal{X}_\Phi)$ still holds. More formally, we demonstrate the following lemma:

**Lemma 4.** *Let us assume there exists a non-empty subset of variables in $\mathcal{X}$ that satisfies Equation 16. Provided no cases of CPMCR exist in $\mathcal{X}$, the following holds: $H(A|\mathcal{X}) = H(A|\mathcal{X}_\Phi)$.*

*Proof.* We let variables $X_j$ and $X_k$ satisfy the property outlined in Equation 16, and show that if removing them to form $\mathcal{X}_\Phi$ leads to $H(A|\mathcal{X}) < H(A|\mathcal{X}_\Phi)$, a contradiction arises. Therefore, it must be true that $H(A|\mathcal{X}) = H(A|\mathcal{X}_\Phi)$. Subsequently, we generalize to the $N$ variable case.

If Equation 16 holds for variables $X_j$ and $X_k$, so does $H(A|\mathcal{X}) = H(A|\mathcal{X}_{\setminus X_j}) = H(A|\mathcal{X}_{\setminus X_k})$. Now we assume that Lemma 4 is not true. In this case, we have: $H(A|\mathcal{X}) < H(A|\mathcal{X}_\Phi)$. Currently, we are only interested in the two-variable case, so this becomes $H(A|\mathcal{X}) < H(A|\mathcal{X}_{\setminus \{X_j, X_k\}})$. Consequently, we have:

$$H(A|\mathcal{X}) = H(A|\mathcal{X}_{\setminus X_j}) = H(A|\mathcal{X}_{\setminus X_k}) < H(A|\mathcal{X}_{\setminus \{X_j, X_k\}}). \tag{30}$$

Equation 30 reveals that in order to verify the condition $H(A|\mathcal{X}) < H(A|\mathcal{X}_\Phi)$, there must be CPMCR, such that $\Psi_{X_j, X_k}(A|\mathcal{X})$ ($\psi$ here is invalid as we are concerned with a single variable). Consequently, a contradiction has arisen; therefore, we have proven Lemma 4 in the two-variable case.

We now generalize this result to the $N$ variable case. Let there be a third variable, $X_l$, that also satisfies the property expressed by Equation 16. In this case, it must be true that: $H(A|\mathcal{X}) = H(A|\mathcal{X}_{\setminus X_j}) =$

$H(A|\mathcal{X}_{\backslash X_k}) = H(A|\mathcal{X}_{\backslash X_l})$. Now we reapply an identical method to demonstrate that, if $H(A|\mathcal{X}) < H(A|\mathcal{X}_\Phi) = H(A|\mathcal{X}_{\backslash\{X_j,X_k,X_l\}})$, we must have a case of CPMCR, such that $\Psi_{X_j,X_k,X_l}(A|\mathcal{X})$. Again, a contradiction has arisen, and this proves the lemma in the case of $N$ variables. $\square$

**Theorem 1.** *Please refer to Section 5.4.*

*Proof.* In Lemma 2, we have shown that variables that provide no information about the actions (satisfying Equation 17) and variables that provide redundant information about the actions (satisfying Equation 16) will not be included in the set $\mathcal{X}_\Phi$. It is therefore trivial to see that the remaining variables in the set $\mathcal{X}_\Phi$ provide non-redundant information about the actions, and, therefore, satisfy $H(A|\mathcal{X}_{\Phi\backslash X_i}) > H(A|\mathcal{X}_\Phi)$. This means that it is not possible to remove any more variables from $\mathcal{X}_\Phi$ without increasing the value of $H(A|\mathcal{X}_\Phi)$. Consequently, the set $\mathcal{X}_\Phi$ satisfies the condition $|\mathcal{X}_\Phi| = \min\limits_{H(A|\mathcal{P})=H(A|\mathcal{X}_\Phi)} |\mathcal{P}|$, where $\mathcal{P} \in \mathscr{P}(\mathcal{X})$. Combining this information with the results of Lemmas 4 and 5, in which we have shown that despite removing these variables the expression $H(A|\mathcal{X}) = H(A|\mathcal{X}_\Phi)$ still holds, completes the proof of Theorem 1. $\square$

# E  Proof of Theorem 2

To carry out this proof we first present Lemma 6, which shows that Algorithm 1 effectively deals with cases of CPMCR between subsets of equal cardinality. Using this result alongside Theorem 1, we demonstrate that the proof is valid under Condition 1.

**Lemma 6.** *Let $\mathcal{P} \in \mathscr{P}(\mathcal{X}_{A_1})$, where $\mathcal{X}_{A_1}$ is generated using Algorithm 1. The following holds[7]:*

$$\forall \mathcal{P} \in \mathscr{P}(\mathcal{X}_{A_1}) \nexists \mathcal{P}' \in \mathscr{P}(\mathcal{X}_{A_1}) : \Psi(A|\mathcal{X}) \quad \& \quad |\mathcal{P}| = |\mathcal{P}'| \quad \& \quad \mathcal{P} \neq \mathcal{P}'. \tag{31}$$

*Proof.* We prove Lemma 6 by demonstrating that Algorithm 1 is able to resolve cases of CPMCR between subsets with equal cardinality. We consider two subsets $\mathcal{P}$ and $\mathcal{P}'$ with equal cardinality ($|\mathcal{P}| = |\mathcal{P}'|$) and CPMCR between them ($\Psi_{\mathcal{P},\mathcal{P}'}(A|\mathcal{X})$). We can write the set $\mathcal{X}$ that contains these subsets as: $\mathcal{X} = \{P_1, P_1', P_2, P_2'...P_N, P_N'\} \cup \mathcal{X}_{\backslash(\mathcal{P},\mathcal{P}')}$, where $P_i \in \mathcal{P}$ and $P_i' \in \mathcal{P}'$. We can now apply Algorithm 1: we begin by calculating $\Phi_{P_1;\mathcal{X} \to A} = 0$, in accordance with Lemma 1. Consequently, $P_1$ gets removed from the set $\mathcal{X}$, so we can write $\mathcal{X} = \mathcal{X}^0_{\backslash P_1}$, where here $\mathcal{X}^0$ indicates $\mathcal{X}$ prior to any algorithmic operations. We now calculate $\Phi_{P_1';\mathcal{X} \to A}$, as we iterate through the set $\mathcal{X}$ ($\mathcal{X}^0_{\backslash P_1}$). To understand the result of this calculation, we first point out that the subsets $\mathcal{P}$ and $\mathcal{P}'$ are defined such that their respective elements combine to provide identical information about the actions (see Equation 2). Therefore, for a single element, the following holds:

$$H(A|\mathcal{X}^0_{\backslash(\mathcal{P},\mathcal{P}')} \cup \mathcal{P}') < H(A|\mathcal{X}^0_{\backslash(\mathcal{P},\mathcal{P}')} \cup \mathcal{P}'_{\backslash P_i'}). \tag{32}$$

Consequently, the following must also be true:

$$\begin{aligned} H(A|\mathcal{X}^0_{\backslash(\mathcal{P},\mathcal{P}')} \cup \mathcal{P}' \cup \mathcal{P}_{\backslash P_1}) &< H(A|\mathcal{X}^0_{\backslash(\mathcal{P},\mathcal{P}')} \cup \mathcal{P}'_{\backslash P_1'} \cup \mathcal{P}_{\backslash P_1}) \\ H(A|\mathcal{X}^0_{\backslash P_1}) &< H(A|\mathcal{X}^0_{\backslash\{P_1,P_1'\}}) \\ H(A|\mathcal{X}) &< H(A|\mathcal{X}_{\backslash P_1'}). \end{aligned} \tag{33}$$

Thus, $\Phi_{P_1';\mathcal{X} \to A} > 0$. We now include $P_1'$ in $\mathcal{X}_{A_1}$ and we do not remove it from $\mathcal{X}$. We will then consider variable $P_2$, and exclude if from $\mathcal{X}$ for similar reasons to $P_1$. The variable $P_2'$ will then remain in the set $\mathcal{X}$ and be added to $\mathcal{X}_{A_1}$ for reasons similar to $P_1'$. This process will be repeated until we have removed all the elements of the subset $\mathcal{P}$ from the set $\mathcal{X}$, while the elements of the set $\mathcal{P}'$ have been added to $\mathcal{X}_{A_1}$. We have therefore proved that Algorithm 1 reliably addresses cases of CPMCR between subsets with equal cardinality. $\square$

**Theorem 2.** *See Section 5.4.*

*Proof.* Given Condition 1 and the proof of Lemma 6 outlined in Appendix E, we can disregard cases of CPMCR, and, therefore, directly apply Theorem 1 to complete this proof. $\square$

---

[7]In this lemma, we adopt the definition of $\Psi(A|\mathcal{X})$ provided in Equation 3.

## F  Tit-For-N-Tats Extra Graphs

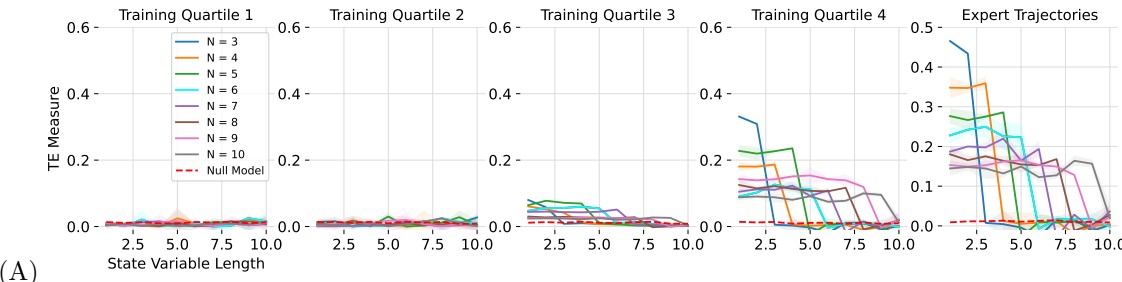

(A)

Figure 11: In subfigure (A), we use $\Phi_{X_i;\mathcal{X}\to A}$ to investigate how entropy is transferred from state variables to actions during different stages of training.

## G  Implementation Details

### G.1  Synthetic Experiments

**Transfer entropy measure estimation.**  For the estimation of the amount of entropy that each variable transfers to the actions, we use a standard feed-forward Neural Network with a ReLu activation function and with one hidden layer with 50 nodes. We select the following values for the parameters in Algorithm 2: $b = 10000$, $N = 20000$, with a learning rate of 0.01.

**Baselines.**  For the calculation of UMFI, we select the exact values of the hyperparameters as those used in (Janssen et al., 2023). As far as PI is concerned, we adopt the following values: $\alpha = 0.01$, and 1000 runs. For the calculation of PI itself, we employed Ridge (Pedregosa et al., 2011), as implemented in Scickit-Learn. In fact, in this case, hyperparameter tuning has no impact on the final UMFI and PI values.

### G.2  The Secret Key Game

**Trajectory generation.**  To carry out the polynomial interpolation in the Secret Key Game, we use the NumPy's `polynomials` package `https://numpy.org/doc/stable/reference/routines.polynomials.package.html`. An expert trajectory for this game is one in which the agent achieves a score of zero. For handling the state spaces of lengths 25 and 50, we employ one-step temporal difference Actor-Critic learning during the execution of the game.

The Actor network is updated using Equation $\theta'_{actor} \leftarrow \theta_{actor} + \alpha_{actor}\gamma^t A\nabla \ln \pi(a^t|s^t, \theta_{actor})$, where $\delta = r + \gamma v(s^t, \theta_{critic}) - v(s^{t+1}, \theta_{critic})$ (Sutton & Barto, 2018). The Critic network is updated using the following equation: $\theta'_{critic} \leftarrow \theta_{critic} + \alpha_{critic}MSE(\delta)$. Both networks are characterized by a single fully connected layer of size 64 with a ReLu activation function. The other values of the hyperparameters of the networks are $\gamma = 0.99, \alpha_{actor} = 0.0001, \alpha_{critic} = 0.001$.

**Transfer entropy measure estimation.**  For the estimation of the amount of entropy that each of the keys transfers to the actions, we again use a network with one hidden layer with 50 nodes and a ReLu activation function. For 25 keys, we use $b = 5000$, $N = 2000$ in Algorithm 2. Instead, for a state with 50 keys, we require the following parameters for Algorithm 2 $b = 10000$, $N = 20000$. For both, we used a learning rate of 0.01.

**Baselines.**  For both 25 and 50 keys, we use the exact hyperparameter values specified in (Janssen et al., 2023) for calculating UMFI, as they are sufficient to identify the correct keys in the game. For the PI calculation, we employ the same hyperparameter values as those used in the synthetic data experiments.

**Evaluation of designed states.** We employ a one-step temporal difference Actor-Critic architecture when evaluating each state. Both networks have a single fully connected layer of size 64 and uses a ReLu activation function. We employ the following values for the parameters: $\gamma = 0.99, \alpha_{actor} = 0.0001, \alpha_{critic} = 0.001$, considering 20,000 episodes of the game (in this game each episode requires only one iteration).

### G.3 OpenAI's Gym: Cart Pole

**Trajectory generation.** We play the game until convergence. An expert trajectory is defined as one that achieves a cumulative reward greater than 475, in line with the criteria proposed by the authors of the environments (Brockman et al., 2016). This is done using the same Actor-Critic architecture as described for the Secret Key Game. We introduce random variables into the state, each following a uniform probability distribution within the range $V_{rand_i} \in [-5, 5]$. However, if the variables are truly random, the specific range should be inconsequential.

**Transfer entropy measure estimation.** We use the same architecture as previously stated for the Secret Key Game, except in this case, we use the following parameters for Algorithm 2 $b = 10000$, $N = 4000$, and $\alpha = 0.01$.

**Baselines.** For these experiments when calculating UMFI we use the same values of hyperparameters as in (Janssen et al., 2023), except for the fact that we adopt 50 trees instead of 100. This was because we saw improved performance using 50 trees for the task at hand. For the PI calculation, we adopted the same values of the hyperparameters as those used for the synthetic data experiments.

**Evaluation of designed states.** When plotting the state performance curves shown in Figure 5, we use the same architecture as for the generation of the trajectories.

### G.4 OpenAI's Gym: Lunar Lander

**Trajectory generation.** In accordance with Open AI's documentation, we consider expert trajectories as those which achieve a cumulative reward above 200. Again, the method is applied after playing the game until convergence. This is done using the same Actor-Critic architecture as described for the Secret Key Game. We inject three random variables into the state, in the same manner as described for the cart-pole game.

**Transfer entropy measure estimation.** We use the same architecture as previously stated, but here we use the following parameters for Algorithm 2: $b = 25000$, $N = 100000$, and a learning rate of 0.01.

**Baselines.** Again, for these experiments when calculating UMFI, we adopt the same values of the hyperparameters as used in (Janssen et al., 2023), except for the fact that we use 50 trees instead of 100 (since 50 trees are sufficient to separate the informative variables from the non-informative ones). For the PI calculation, we again adopted the same values of the hyperparameters as used for the synthetic data experiments.

**Evaluation of designed states.** For the evaluation of the training times corresponding to states composed of different numbers of variables, we use the same architecture as that used for generating trajectories. In this case, we only run the experiment for 3000 episodes.

### G.5 OpenAI's Gym: Pendulum

**Trajectory generation.** Again we play the game until convergence. This is done using a PPO architecture, with the following Actor update rule:

$$\theta' \leftarrow \theta + \alpha\gamma^t \nabla min(\frac{\pi(a^t|s^t, \theta^t)}{\pi(a^t|s^t, \theta_k^t)} A^t, clip(\frac{\pi(a^t|s^t, \theta^t)}{\pi(a^t|s^t, \theta_k^t)}, 1 - \epsilon, 1 + \epsilon) A^t, \tag{34}$$

where the surrogate actor had its weights updated as described except it is multiplied by 0.95. For this algorithm, we use the following values for the hyperparameters: $\gamma = 0.99, \alpha = 0.0003, \epsilon = 0.2$. We also assume a mini-batch size of 64 and update our networks every 2048 steps. We use 10 epochs and an entropy coefficient of 0.001. The network has one hidden layer with 64 nodes and uses continuous hyperbolic tangent activation functions according to Schulman et al. (2017). We consider trajectories with cumulative rewards greater than -10 as expert trajectories.

**Transfer entropy measure estimation.** We use the same architecture to estimate the TE values, combined with the following parameters for Algorithm 2 $b = 5000$, $N = 4000$, with a learning rate of 0.01.

**Baselines.** For these experiments, when calculating UMFI we employ the same values of the hyperparameters as those used in (Janssen et al., 2023). For the PI calculation, we adopted the same hyperparameters as used for the synthetic data experiments.

**Evaluation of designed states.** We use the same PPO architecture when investigating the performance of different states as that used for generating the trajectories.

### G.6 TFNT in the Iterated Prisoner's Dilemma

**Player 1**

|  |  | Cooperate | Defect |
|---|---|---|---|
| **Player 2** | Cooperate | 2, 2 | 3, 0 |
|  | Defect | 0, 3 | 1, 1 |

Figure 12: Iterated Prisoner's Dilemma payoff matrix.

**Trajectory generation.** We employ tabular Q-learning where $\alpha = 0.9$, and the value of $\epsilon$ decays from 1 to 0 during $40,000$ iterations ($\epsilon^t = \epsilon^{t-1} - (1/40,000)$), while $\gamma$ is equal to 0.99. We play the game with the above payoff matrix until convergence. We use one-shot encoding in our implementation to generate the state at a certain time step. For example, let us assume that the last 9 action pairs could be represented by $s^t = [(C,D),(C,D),(C,C),(D,D),(C,D),(D,D),(C,D),(D,C),(D,C)]$, this can be re-written as: $s^t = [(2),(2),(0),(3),(2),(3),(2),(1),(1)]$.

**Transfer entropy measure estimation.** We use the same architecture as for the other experiments combined with the following values of the parameters for Algorithm 2: $b = 1000$, $N = 10000$, with a learning rate of 0.01.

**Baselines.** For these experiments, for the calculation of UMFI, we use the exact values of the hyperparameters as those in (Janssen et al., 2023). We select these values since hyperparameter tuning has no impact on the final UMFI values. For the PI calculation, we again adopt the same hyperparameters as those used for the synthetic data experiments.

**Evaluation of designed states.** We employ the same Q-learning architecture when evaluating the training performance of different state representations.

### G.7 Null Model

The null model consists of a series of random integers between 0 and 1 that are added to the trajectories after their generation. For all experiments, the values of the hyperparameters used in Algorithm 2 to estimate $\Phi_{NM;\mathcal{X} \to A}$ are the same as for the variables of interest. For a variable to be considered as influencing the actions in a statistically significant way they must have a 95% chance of falling outside the range of the null model, and, therefore, $\mu_{Xi} - \frac{2\sigma_{Xi}}{\sqrt{10}} > \mu_{NM} + \frac{2\sigma_{NM}}{\sqrt{10}}$, for all variables included in $\mathcal{X}_*$.

