# OpenReview forum: "Information-Theoretic State Variable Selection for Reinforcement Learning"
_TMLR — Rejected by TMLR_

### Review · Reviewer_TWmq · 2025-06-19

**Summary Of Contributions:**

The authors propose a new method, based on the transfer entropy (TE), to select a subset $ X^* $ of state features $ X = \\{X_1, \dots, X_N\\} $ that are useful to a given policy $ \pi $ (in practice, a near-optimal policy $ \pi^* $ derived using reinforcement learning). Under some assumptions, this method is proved to select one of the smallest subset of features $ X^*  \subseteq X $ such that $ H(A | X^*) = H(A | X)$. They empirically validate the effectiveness of this method compared to other methods.

**Audience:**

Yes

**Broader Impact Concerns:**

No concerns.

**Claims And Evidence:**

No

**Requested Changes:**

### Claims and discussions

- Could you answer my concerns about the supposed "weakness" of the required conditions for Theorem 2?
- Could you better motivate why the method is useful despite the fact that it requires to know an optimal policy?
- Could you fix the definitions of equations (2) and (3) and better explain how these and "Condition 1" makes your algorithm correct?
- Could you better discuss eventual limitations of the method?

### Minor

- It may be worth explaining (in Section 3.2 or after) whatis $ X^t $, what is $ X^{t-1} $ and what is $ Y^{t-1} $ in your case.

**Strengths And Weaknesses:**

### Strengths

- The method improves on previous methods, in the sense that it selects the optimal subset even in the presence of redundant subsets (under some assumptions).
- The idea of using a random variable as a null model is a great idea as an alternative to arbitrary thresholds.
- The paper is citing related works extensively (even though I think that the related methods could sometimes be explained a little bit more).
- There are relatively few typos and few unclear notations (see the ones highlighted below).

### Weaknesses

- No discussion of the limitations of the methods, and while the assumptions (conditions) are claimed "weak", I am not sure to agree.
    - You state that the assumptions are not strong, but I think that the method could fail in simple cases. One example I can think of is $ x = [r, \theta, y] $ with $ x^* = {y}$ and $y = r \sin(\theta)$. In this case, "Condition 1" is not satisfied, and the method fails if we select the variable $ y $ first in your algorithm. So is "Condition 1" really a weak assumption? If yes, could you better justify this?
- Not a clear motivation on the usefulness of such methods.
    - You state that removing features could speed up learning, but we need to know an optimal policy for applying your method. Why is it useful to select the smallest subset $ X^* $ of relevant variables for the optimal policy if the process for obtaining such variables already requires to know an optimal policy conditioned on the full state $ X $? Could you give some use cases of this subset when we already know an optimal policy?
    - (Maybe irrelevant) You say that some methods fail to provide to reduce the state dimensionality, but do we always want to remove state variables instead of computing (smaller or bigger) features? (e.g., compute $ x^* = r \sin(\theta) $ from $ x = [r, \theta] $). Maybe this is irrelevant since computing useful representations is a different task than selecting features.
-  The distribution $ p(x, a) $ with $ x = (x_1, \dots, x_N) $ under which the mutual information (and thus TE) is evaluated is not so clearly defined (in my opinion, the last paragraph of section 3.1 should be improved). Do you have a finite horizon $ H $ and you define $ p(x, a) = \frac{1}{H} \sum_{t=0}^{H - 1} p(X_t = x, A_t = a) $? Note that from this joint distribution over all considered random variables, all marginal distributions and conditional distributions are defined. (There may be other valid distributions to consider of course.)
- The conditions are not so easy to read and understand. It makes it difficult to understand your theoretical results.
    - Why is $ \psi_{\mathcal{P}'} $ a condition defined for a particular partial set $ \mathcal{P}' $, while for $ \Psi_{\mathcal{P}, \mathcal{P}'}(Y | X) $, the subscript $ (\mathcal{P}, \mathcal{P}') $ does not denote any particular set apparently, since you use existence quantifiers $ \exists $ in equation (3).
    - The subset $ \mathcal{P} $ is not defined in equation (2). Is there a universal quantifier $ \forall \mathcal{P} $ missing?
    - In equation (3), in the definition of $ \mathcal{P}' $, do you mean that $ \mathcal{P} $ and $ \mathcal{P}' $ are disjoint?
- The presentation of the empirical results could be improved.
    - I am sorry if it should be obvious, but I do not understand the y-axis in Figure 1. TERC is a criterion, not a value, right? What does the approximate value of 0.35 represents and why is this value comparable in the case of TERC and UMFI?

### Minor

- Some parts could be clarified. In particular, I did not understand the explanation of TERC given in the second paragraph of 5.4 nor the explanations of the conditions and CPMCR given in section 3.3.
- In the presentation of the secret key game, could you clarify the following points.
    - When you write several times $ v_\text{rand} $, shouldn't you use $ v_{\text{rand}_i} $ or something similar to highlight the fact that it is not the same value at every position in the vector? Or is it?
    - Are the secret keys kept constant over a complete episode? In other words, in a given episode are $v_\text{rand}$ variables the only parts of the state that change? And if yes, are the secret keys different at each episode?

### Questions

- Is the set size $ | X^* | $ a good measure of the complexity of an input for a policy? Do we really want to find the subset whose size is minimum ? Wouldn't information-theroetical metrics such as the entropy be interesting also? For example, with $ X = [X_1 = 100 X_2 + 10 X_3 + Y, X_2 = U([0, 9]), X_3 = U([0, 9])] $ with $ Y =  U([0, 9]) $, if $Y$ is irrelevant, wouldn't it be better to select $ X_2 $ and $ X_3 $ instead of $ X_1 $?
- I do not understand how a "greedy" method could select the optimal subset of features. Intuitively, an optimal method would have an combinatorial complexity because it would need to test every possible subset of $ X $. Am I wrong here? Is it the case without additional assumptions? If "Condition 1" is the reason why your greedy method is is working, it raises again the concern that it may be a pretty strong assumption. Do you have an intuitive explanation of why the greedy method works in this case?
- In section 3.1., why considering deterministic transitions and rewards. Do you need this assumption? Moreover, what is the notation $ a^t(s^t) \in \mathcal{A} $? Finally it is not so clear throughout the paper whether $\pi(a^t | s^t)$ is a deterministic or stochastic policy. In 6.4.1, you write $ \pi (a^t | s^t): \mathcal{S} \mapsto \mathcal{A} $.
- Theorem 2: are you sure that your conditions exclude the fact that there exists several optimal subsets? If not, the conclusion of the theorem $ C_1 \Rightarrow X_* = X_{A_1} $ should rather be $ C_1 \Rightarrow | X_* | = | X_{A_1} | $, or am I mistaken?

---

> ### Author Response · Authors · 2025-07-11
> **Reviewer Response**
>
> #Comment 1
> “You state that the assumptions are not strong, but I think that the method could fail in simple cases.”
>
> ### Response
>
> In this case, the method can be seen as “failing,” but the failure is not severe. TERC would select ${r,\theta}$, which is still a meaningful and effective representation, just not the most succinct one described using Equation 5. We have clarified in the manuscript that even when assumption is not satisfied, our method still selects an optimal performing set, thought it may not be minimal.
>
> #Comment 2
>
> “You state that removing features could speed up learning, but we need to know an optimal policy for applying your method?”
>
> ### Response
>
> Reducing the number of variables after training offers practical benefits at deployment time, including lower resource consumption and faster inference. We aim to highlight this in the introduction with the following sentences:
>
> “Retaining such variables can result in unnecessary computations during deployment\footnote{Indeed, this is of critical importance given the increasing energy footprint of today's machine learning systems \citep{patterson_2022_carbon}.}.”
>
> #Comment 3
>
> “The distribution $p(x,a)$ with $x=(x_1,\elipses,x_N)$ ...”
>
> ### Response
>
> At present, we consider the distribution as defined with a horizon length of 1. The notation \( x = (x_1, \ldots, x_N) \) does not appear in the paper. Instead, we use the notation \( s^t = [x^t_1, x^2_t, \ldots, x^t_N] \) to indicate that a realization of the state at time \( t \) is a vector composed of individual state variables. This is intended to emphasize that the state is structured as a collection of components, each corresponding to a distinct variable.
>
> #Comment 4
> “Why is $\psi_{\mathcal{P}'}$ a condition...”
>
> ### Response
> We agree and would consequently rewrite $\Phi_{\mathcal{P}’,\mathcal{P}}$ as $\Phi$ in an updated version of the manuscript.
>
> #Comment 5
>
> “The subset $\mathcal{P}$ is not defined in equation (2). Is there a universal quantifier $\forall \mathcal{P}$  missing?”
>
> ### Response
>
> We would like to thank the reviewer to point this out. In the updated version of the manuscript, we substitute the following sentence:
>
>  “CPMCR will be defined as a condition that, if true, signifies the existence of two subsets ($\mathcal{P}$ or $\mathcal{P}'$) that convey identical information about the target.”
>
> With the following:
>
>  “CPMCR will be defined as a condition that, if true, signifies the existence of two subsets ($\mathcal{P}, mathcal{P}' \in \mathscr{P}(\mathcal{X}): \mathcal{P}\neq mathcal{P}$) that convey identical information about the target.”
>
> #Comment 6
> “I am sorry if it should be obvious, but I do not understand the y-axis in Figure 1.”
>
> ### Response
>
> Indeed, TERC is a condition, whereas the quantity we are measuring in these graphs is \( \Phi_{X_i;\mathcal{X} \rightarrow A} \). Would the reviewer find it helpful if we updated the graph labels to reflect this distinction more explicitly? We are happy to make this change if it would improve clarity and readability.
>
> #Comment 7
>
> “When you write several times $v_{rand}$, shouldn't you use $v_{rand}_i$ or something similar to highlight the fact that it is not the same value at every position in the vector? Or is it?”
>
> ### Response
>
> The reviewer is indeed correct; it is not the same value at every key. We have updated the manuscript to include $v_{rand}_i$.
>
> #Comment 8
>
> “Are the secret keys kept constant over a complete episode?”
>
> ### Response
>
> Thanks for the request for clarification: the keys change every iteration of every episode.  We have clarified this point in the revised version of the manuscript.
>
> #Comment 9
>
> “Is the set size $|\mathcal{X}*|$ a good measure of the complexity of an input for a policy?”
>
> ### Response
>
> We thank the author for this interesting point. They are indeed correct that it may be more sample efficient to learn from a state that aims to optimize for entropy. However, more state variables will lead to greater compute required at the time of deployment. And, as the reviewer demonstrated, optimizing for entropy can lead to a larger state representation.
>
> #Comment 10
>
> “I do not understand how a "greedy" method could select the optimal subset of features.”
>
> ### Response
>
> That is an excellent point and it underscores a core benefit of our approach. Exhaustive search across all subsets is only required when insight arises from variables acting together—what we refer to as synergistic information. Our method uncovers these interactions in linear time by starting with the full variable set, then pruning any variable that does not contribute additional information when evaluated in concert with the rest.
>
> #Comment 11
>
> “In section 3.1., why considering deterministic transitions and rewards.”
>
> ### Response
>
> We do indeed consider stochastic policies, we have updated our notation too: $
> \pi_{\text{sto}} : \mathcal{S} \times \mathcal{A} \to [0,1]$. Thanks for spotting this issue.

---

> > ### Comment · Reviewer_TWmq · 2025-07-12
> > **Thank you for your responses**
> >
> > Dear Authors,
> >
> > **Comment 1.** I agree that it is interesting that your method still selects an effective representation when C1 is not satisfied, it is nice that you pointed it out in the conclusion. But note that your methods fails to selects a minimal subset (which is normal when C1 is not satisfied). My concern was not that it fails when C1 is not satisfied, but that C1 is not satisfied for such a simple case. As a conclusion, I am really not sure C1 is a week assumption, and I have the feeling that the strength of this assumption is what makes a greedy algorithm able to solve this combinatorial problem.
> >
> > In any case (whether this is a strong or weak assumption), I think that it would be more honnest to always state that your method obtains an optimal and minimal subset *under conditions on the set of state features* (in the introduction notably). For example, the following claim is, in my opinion, too strong: "In practical terms, we will introduce a set of methods that will allow us to derive the minimal and optimal set of variable for state representation in presence of redundant and synergistic relationships".
> >
> > **Comment 2.** That does not address my concern: if your methods requires to learn/know an optimal policy, it should be clearly stated, already in the introduction. That said, it is good that you motivated why the method can be useful, even when an optimal policy is known (e.g., to learn a second policy and reduce the number of state variables and computation).
> >
> > **Comment 3.** I do not understand your answer. The last paragraph of subsection 3.1 is still unclear to me. Measuring the conditional entropy of random variables implicitly makes the assumption of an underlying conditional/joint distribution for these random variable. Could you explain the considered joint distribution of random variables? You answered that you use an horizon of "1", I do not understand. It would mean that the distribution over $s$ that you select is the initial state distribution. I suppose that you do not do that. So, how to you aggregate the timesteps?
> >
> > **Comment 4.** Ok.
> >
> > **Comment 5.** If it is the case, then the criterion should not have a subscript $ \mathcal{P} $, and there should be two additional existence quantifiers in equation (2) for both $ \mathcal{P} $ and $ \mathcal{P}' $, right? Maybe I am misunderstanding something here.
> >
> > **Comment 6.** Yes, the values that you report in the graph and discuss below should be explained somewhere. Maybe the explanation was present in the text and I missed it. If not, I think that writing it on the y-axis would be great.
> >
> > **Comment 7.** Ok.
> >
> > **Comment 8.** By "every iteration of every episode", do you mean every episode? Not every timestep of every episode, right?
> >
> > **Comments 9.** Ok.
> >
> > **Comments 10.** Ok.
> >
> > **Comments 11.** Ok.
> >
> > Best regards, \
> > Reviewer TWmq

---

> > > ### Author Response · Authors · 2025-07-12
> > > **Reviewer response**
> > >
> > > ### Comment 1
> > >
> > > > In any case (whether this is a strong or weak assumption), I think that it would be more honnest to always state that your method obtains an optimal and minimal subset under conditions on the set of state features (in the introduction notably).
> > >
> > > **Response**
> > > We update
> > >
> > > > “In practical terms, we will introduce a set of methods that will allow us to derive the minimal and optimal set of variable for state representation in presence of redundant and synergistic relationships.”
> > >
> > > to become
> > >
> > > > “In practical terms, we will introduce a set of methods that, in conjunction with the satisfaction of an assumption, will allow us to derive the minimal and optimal set of variable for state representation in presence of redundant and synergistic relationships. Even when this assumption does not hold, our method still identifies an optimal, though not necessarily minimal, representation.”
> > >
> > > ---
> > >
> > > ### Comment 2
> > >
> > > > That does not address my concern: if your methods requires to learn/know an optimal policy, it should be clearly stated, already in the introduction.
> > >
> > > **Response**
> > > We agree. The following
> > >
> > > > “If this value is bigger than zero, the actions are said to depend on this state variable, and TERC is verified.”
> > >
> > > will become
> > >
> > > > “If this value is bigger than zero, the actions are said to depend on this state variable, and TERC is verified. Under these circumstances, the state variable is considered informative and included in our representation. Once all the informative state variables have been identified, our agent can be re-trained on the lightweight state, leading to greater efficiency at the time of deployment.”
> > >
> > > ---
> > >
> > > ### Comment 3
> > >
> > > > “I do not understand your answer. The last paragraph of subsection 3.1 is still unclear to me. Measuring the conditional entropy of random variables implicitly makes the assumption of an underlying conditional/joint distribution for these random variable. Could you explain the considered joint distribution of random variables? You answered that you use an horizon of ‘1’, I do not understand. It would mean that the distribution over $s$ that you select is the initial state distribution. I suppose that you do not do that. So, how do you aggregate the timesteps?”
> > >
> > > **Response**
> > > We consider the full range of training data for our joint and marginal distributions. For example, the quantity
> > >
> > > $$
> > > \Phi_{X_i;\, \mathcal{X} \rightarrow A}
> > > $$
> > >
> > > measures the information flow from a particular state variable $X_i$ (conditioned on the remaining state variables $\mathcal{X} \setminus \{X_i\}$) to the actions $A$ at time $t$. To make this concrete, suppose we observed the following training trajectory:
> > >
> > > $$
> > > (s^{t=1} = [x_0^{\,t=1}, x_1^{\,t=1}],\; a^{t=1},\; s^{t=2} = [x_0^{\,t=2}, x_1^{\,t=2}],\; a^{t=2},\; s^{t=3} = [x_0^{\,t=3}, x_1^{\,t=3}],\; a^{t=3})
> > > $$
> > >
> > > If we are computing the mutual information $I(\mathcal{X}; A)$ at time $t$, we require the joint distribution $P(s^t, a^t)$ and the product of marginals $P(s^t)\,P(a^t)$, where this distribution is defined over all three samples. We hope this clarifies our approach.
> > >
> > > ---
> > >
> > > ### Comment 4
> > >
> > > > If it is the case, then the criterion should not have a subscript $\mathcal{P}$, and there should be two additional existence quantifiers in equation (2) for both $\mathcal{P}$ and $\mathcal{P}'$, right? Maybe I am misunderstanding something here.
> > >
> > > **Response**
> > > Could it be that the confusion is arising because we did not explicitly state that $\mathcal{P} \neq \mathcal{P}'$? When writing equation 5, it then becomes clear that both of these subsets satisfy this condition.
> > >
> > > ---
> > >
> > > ### Comment 5
> > >
> > > > Yes, the values that you report in the graph and discuss below should be explained somewhere. Maybe the explanation was present in the text and I missed it. If not, I think that writing it on the y-axis would be great.
> > >
> > > **Response**
> > > We will update this in the paper.
> > >
> > > ---
> > >
> > > ### Comment 6
> > >
> > > > By “every iteration of every episode”, do you mean every episode? Not every timestep of every episode, right?
> > >
> > > **Response**
> > > Apologies for the unclear language. We change the key every iteration, and arbitrarily set an episode as one iteration long.

---

> > > > ### Comment · Reviewer_TWmq · 2025-07-12
> > > > **Thank you for the clarifications**
> > > >
> > > > Dear Authors,
> > > >
> > > > Thank you for the clarifications, I think that the changes you proposed will help the reader better understand the scope of the contributions.
> > > >
> > > > Regarding comment 4, sorry for the confusion, I had forgot the meaning of equation (2). After reading equation (2) and (3) again, everything is ok for me.
> > > >
> > > > I just have one follow-up question concerning the distribution, because it is still a bit unclear to me. From the beginning of your answer, I understand that you compute a different mutual information at every timestep. But at the end, you say that the distribution is taken over all three timesteps. According to this, it means that you implicitly use the distribution $p(S = s) = 1/H \sum_{t=1}^{H} p(S_t = s)$ with $H = 3$.  Is that correct ? This is what I meant in my original review (note that the horizon $H$ does not need to be fixed, it could be considered a random variable instead).
> > > >
> > > > If yes, I think this is indeed the most natural choice (you consider the marginal distribution over all states/actions over all timesteps), but I just think that it is not so clearly formalized in the end of subsection 3.1. I would suggest to make it a little more clear in that paragraph.
> > > >
> > > > Kind regards, \
> > > > Reviewer TWmq

---

> > > > > ### Author Response · Authors · 2025-07-12
> > > > > **Reviewer Response**
> > > > >
> > > > > **Comment**
> > > > >
> > > > > I just have one follow-up question concerning the distribution, because it is still a bit unclear to me. From the beginning of your answer, I understand that you compute a different mutual information at every timestep. But at the end, you say that the distribution is taken over all three timesteps. According to this, it means that you implicitly use the distribution
> > > > >
> > > > > $$
> > > > > p(S = s) \;=\; \frac{1}{H} \sum_{t=1}^{H} p(S_t = s)
> > > > > $$
> > > > >
> > > > > with \(H = 3\). Is that correct? This is what I meant in my original review (note that the horizon \(H\) does not need to be fixed; it could be considered a random variable instead).
> > > > >
> > > > > ---
> > > > >
> > > > > **Review**
> > > > >
> > > > > Sorry for the confusion, yes, your definition is correct and was all along. Initially, I misunderstood—I thought you were assuming we were calculating \(I(\mathcal{X}^{t;t-n};A)\) because of the phrase “The distribution \(p(x,a)\) with \(x=(x_1,\ldots,x_N)\).”
> > > > >
> > > > > The sentence:
> > > > >
> > > > > We also sample from \(s^{t} = [x^{t}_{1}, x^{t}_{2}, \ldots, x^{t}_{N}]\) to derive the set of random variables \(\mathcal{X} = \{X_{1}, X_{2}, \ldots, X_{N}\}\). In order to derive the random variables associated with the agent’s actions, we follow an identical sampling process, leading to \(A\).
> > > > >
> > > > > will become:
> > > > >
> > > > > We also sample from \(s^{t} = [x^{t}_{1}, x^{t}_{2}, \ldots, x^{t}_{N}]\) to derive the distribution set of random variables \(\mathcal{X} = \{X_{1}, X_{2}, \ldots, X_{N}\}\), such that \[p(\mathcal{X}=s) \;=\; \frac{1}{T}\sum_{i=1}^{T} p(\mathcal{X}^{t}=s).\] In order to derive the random variables associated with the agent’s actions, we follow an identical sampling process, leading to \(A\), where \[p(A=a) \;=\; \frac{1}{T}\sum_{i=1}^{T} p(A^{t}=a).\]

---

### Review · Reviewer_E9be · 2025-06-27

**Summary Of Contributions:**

This paper proposes Transfer Entropy Redundancy Criterion (TERC), an information-theoretic quantity which measures how much state variables influence the actions taken by a trained agent. They demonstrate (using both a naive procedure and one more suited for practice) how this criterion can be used to generate an approximate minimal set of variables which influence the actions.

They present experiments on a range of synthetic environments, classic small-scale RL environments, and in an iterated prisoner's dilemma environment. They compare against the Ultra Marginal Feature Importance (UMFI) and Permutation Importance (PI) algorithms.

**Audience:**

Yes

**Claims And Evidence:**

Yes

**Requested Changes:**

- A couple of the figures are difficult to read (bar labels in particular). Can these be made more clear?

**Strengths And Weaknesses:**

**Strengths**
- This is an important problem, and the information-theoretic approach taken is novel and interesting.
- The theoretical results appear correct to my knowledge, with the proofs clearly written and explained in the appendix.
- The proposed technique outperforms both baselines across the experimental suite.


**Weaknesses**
- Perhaps the largest weakness is that the method requires expert trajectories/running an RL agent to convergence (at least as far as I understand the method/the descriptions of the experiments). One would hope that this method can be run iteratively with training an RL agent to convergence, in order to save computation time if a state dimension can be identified as spurious early on in training. Despite this, I do not think this is a fatal flaw and can potentially be highlighted as future work.

---

> ### Author Response · Authors · 2025-07-11
> **Reviewer Response**
>
> ### Comment 1
> “A couple of the figures are difficult to read (bar labels in particular). Can these be made more clear?”
>
> **Response**
>
> Thank you for highlighting this issue. In the revised manuscript we enlarged the bar charts and increased the font size of their axis labels and tick marks to ensure readability.

---

### Review · Reviewer_FMbR · 2025-06-28

**Summary Of Contributions:**

This paper proposed a method for state representation dimensionality based on information theoretic notions. The method attempts to remove redundant and noninformative state variables that do not affect the action of a policy.

**Audience:**

Yes

**Claims And Evidence:**

No

**Requested Changes:**

See major / minor comments above.

**Strengths And Weaknesses:**

Major:
- In the introduction, it is claimed that the “​​uninformative variables can impede the learning process of iterative decision-making systems” but the proposed method is intended to be applied after training. This seems to be a major shortcoming of the paper. Although a simpler state representation can be good for inference time (after training), it seems to me that the primary benefits might come from improved training efficiency and stability. Can the authors give further justification for this?

- Can this method be applied to any “behavioral policy”? If so, one suggestion I have is to redo the setup to focus on a setting where there exists offline / behavioral data. The way it is currently formulated to be “after training” seems a bit strange to me.

- Relatedly, what if the initial training process led to a bad policy? Consider a situation where the training process ends in a constant policy. Then state variable selection would remove all states. Now we are left with no states and if we retrain a policy, it won’t learn anything.
Similarly, consider a situation where we are in a high-dim state space with most dimensions being noise. The initial training process may learn some spurious correlations between noise and actions. How would this approach work in these extreme situations?

- Apologies if I missed it, but I don’t think the empirical results consider the total training cost (training on full state, performing state dim reduction, training on reduced state). Is this correct? What is missing seems to be a systematic study that compares:
  - Baseline: One long continuous training on the full state
  - New Approach: A two step process (training on full state, state reduction, training on reduced state). The cost of the entire two stage process should be combined and compared to the baseline.


Minor:
- In section 3.1, missing comma in definition of s_t and also \mathcal X.
- In section 3.1, the definition of p_X and \mathcal X could be written more precisely, I think. Might be more precise to say that X is sampling from the empirical distribution of x in the training trajectory?
- Section 3.3 is hard to follow. The authors immediately say “...existence of two subsets P or P’ that convey identical information about the target…” But subsets of what? What is the target? I feel like this section could be significantly clarified to help the reader understand.
- I think Section 4 might benefit from coming before all of the technical definitions in Section 3.
- I could not easily parse Eq (5). Couldn’t we write this more simply? The way it is written is confusing and unintuitive.
X_* \in \argmin_{P: H(A|P) = H(A|X) } |P|
- I did not follow the sentence below. Convergence of what?
  - The formulation presented in Equation 5 possesses desirable properties, such as guaranteed convergence in line with Theorem 4(1) in (Li et al., 2006).
- How Bayesian networks are used in the approach should be clarified and written down precisely.

---

> ### Author Response · Authors · 2025-07-11
> **Reviewer response**
>
> ### Comment 1
> “In the introduction, it is claimed …”
>
> **Response**
>
> We would like to thank the reviewer for this comment. Indeed, reducing the number of state variables also allows rapid and efficient inference at deployment. We clarified this point by adding the following sentences to the introduction: Failure to remove such variables will lead to unnecessary computations once deployed. This is critically important given the increasing energy footprint of today’s machine-learning systems (Patterson et al., 2022).
>
> ---
>
> ### Comment 2
> “Can this method be applied to any ‘behavioral policy’?”
>
> **Response**
>
> Thank you for this insightful comment. Yes, the proposed method is applicable to *any* behavioural policy. Our formulation concentrates on the post-training phase because, during deployment, reinforcement-learning systems often perform excessive computations on superfluous state variables. Your suggestion to reframe the setup around a more general offline or behavioural-data setting is well taken; it would integrate naturally with standard offline RL paradigms and could broaden the applicability of our approach. While we do not explore this direction in the current work, we acknowledge its importance and plan to investigate it in future work.
>
> ---
>
> ### Comment 3
> “Relatedly, what if the initial training process led to a bad policy?”
>
> **Response**
>
> We believe that this statement overlooks an importance nuance. If we learn a bad policy, we will select the state variables required to learn this bad policy. This may mean that upon re-learning we restrict ourselves to only bad policies, which is indeed a disadvantage.
>
> ---
>
> ### Comment 4
> “Apologies if I missed it …”
>
> **Response**
>
> We thank the reviewer for their thoughtful suggestion and would appreciate some clarification to ensure we address the point accurately. Specifically, are you proposing a comparison between:
>
> 1. Training on the full state for **2000** episodes, *versus*
> 2. Training on the full state for **1000** episodes, applying our state-reduction method, and continuing training on the reduced state for another **1000** episodes?
>
> We would be happy to include such an analysis if this aligns with your intent.
>
> ---
>
> ### Comment 5
> “In Section 3.1, missing comma …”
>
> **Response**
>
> Thank you for pointing this out; the comma has been added in the revised manuscript.
>
> ---
>
> ### Comment 6
> “In Section 3.1, the definition of \(p_X\) …”
>
> **Response**
>
> We agree. We replaced
>
> > “We denote the set of the random variables obtained in this way by \(X = \{X_1, X_2, \dots, X_N\}.\)”
>
> with
>
> > “We also sample from \(s^{t} = [x^{t}_{1}, x^{t}_{2}, \dots, x^{t}_{N}]\) to derive the set of random variables \(X = \{X_1, X_2, \dots, X_N\}.\)”
>
> ---
>
> ### Comment 7
> “Section 3.3 is hard to follow.”
>
> **Response**
>
> We have replaced
>
> > “CPMCR will be defined as a condition that, if true, signifies the existence of two subsets (\(\mathcal{P}\) or \(\mathcal{P}'\)) that convey identical information about the target.”
>
> with
>
> > “CPMCR is a condition that, if satisfied, indicates the existence of two distinct subsets \(\mathcal{P}, \mathcal{P}' \in \mathscr{P}(\mathcal{X})\) such that \(\mathcal{P} \neq \mathcal{P}'\) and both convey identical information about the target.”
>
> ---
>
> ### Comment 8
> “The formulation presented in Equation 5 possesses desirable properties, such as guaranteed convergence in line with Theorem 4(1) in Li et al. (2006).”
>
> **Response**
>
> The sentence now reads:
>
> > “Because the state variables specified in Equation 5 convey the same information as the full set, they inherit the convergence guarantees established by Theorem 4(1) of Li et al. (2006).”

---

> > ### Comment · Reviewer_FMbR · 2025-07-29
> >
> > Thanks for your responses.
> >
> > > We believe that this statement overlooks an importance nuance. If we learn a bad policy, we will select the state variables required to learn this bad policy. This may mean that upon re-learning we restrict ourselves to only bad policies, which is indeed a disadvantage.
> >
> > It is now clearer that you are assuming that we start with a fixed policy and then try to learn a smaller set of state variables. Essentially it is policy distillation, I think. But the paper is written in a way that makes it quite easy to misinterpret. For example, in the abstract, you say that this works for Q-learning, AC, PPO, etc, but if we assume that we a fixed policy, why does it matter where it came from?
> >
> > The abstract also says "resulting in more sample-efficient learning", but this is also misleading because the learning has already happened and now the focus is on distillation, not learning.
> >
> > > We thank the reviewer for their thoughtful suggestion and would appreciate some clarification to ensure we address the point accurately. Specifically, are you proposing a comparison between:
> >
> > Yes, this what I was suggesting, but I suggested it because I thought the paper is focusing on "learning". If the authors want to focus on learning, then this is the route I suggest they take going forward (i.e., by formulating the true task as learning, and then splitting up samples across learning & distillation phases--I believe that this is the right way to look at this problem). If the authors want to only focus on distillation, then they need to make it crystal clear throughout the paper.

---

> > > ### Author Response · Authors · 2025-07-29
> > > **Response**
> > >
> > > # Response
> > >
> > > We agree with the reviewer that we were not clear enough that this is indeed policy distillation.
> > >
> > > Consequently, in the abstract, "We define an algorithm based on TERC that provably excludes variables from the state that have no effect on the final performance of the agent, resulting in more sample-efficient learning. Experimental results show that this speed-up is present across three different algorithm classes (represented by tabular Q-learning, Actor-Critic, and Proximal Policy Optimization (PPO)) in a variety of environments." Will be re-written as "\textcolor{red}{We define an algorithm based on TERC that provably excludes variables from the state that do not affect the agent's policy during learning, resulting in more efficient inference. Our approach is policy-dependent, making it agnostic to the underlying learning paradigm. Consequently, we use our method to enhance efficiency across three different algorithm classes (represented by tabular Q-learning, Actor-Critic, and Proximal Policy Optimization (PPO)) in a variety of environments.}"
> > >
> > > Furthermore, in the introduction "More specifically, TERC is based on the quantification of the reduction in uncertainty of the realizations of the policy when considering the set of state variables with and without the variable under consideration." Will become "More specifically, TERC is based on the quantification of the reduction in uncertainty of the realizations of the policy when considering the set of state variables with and without the variable under consideration. \textcolor{red}{This makes TERC a policy-dependent method, while being agnostic to the underlying learning algorithm.}"

---

### Decision · Action_Editor_dcmB · 2025-08-17

**Recommendation:** Reject

**Additional Comments:**

The paper needs substantial improvement in writing, even though the revised version has made good efforts and clarified a few important misunderstandings. Many key quantities are undefined (unless I missed it). Examples are $\mathscr{P}(X)$, $\mathcal{X}_{\setminus \mathcal{P}}$, $\mathcal{X}_{\setminus (\mathcal{P} \cup \mathcal{P}‘} \cup \mathcal{P}’_{P’}$. Readers may be able to guess some of them, but there will be misunderstanding and confusion with such heavy and complex notation. Furthermore, the authors may consider improving the structure and flow of the paper, and consolidating/shortening the writing quite a bit. For example, instead of formalizing the problem in section 4 on page 6, it’d be better to move it up earlier for greater clarify, and Related Work might be moved to a later section. See below for a few more detailed questions about writing.

Page 5: What is an “incomplete subset”?
Page 5: Critical but undefined notation throughout the section.
Page 8: Lemma 1 and Theorem 1 are poorly worded. We may say “Given condition C, we have result R”, or “We have: C ==> R”, but not “Given condition C, we have: C ==> R”.
Page 9: Section 5.5 doesn’t explain clearly how the approaches work.
Page 9: In experiments, it’s helpful to investigate stability of TERC. That is, if you re-run policy learning (say, with different seeds or params), will TERC produce different results?

**Audience:**

Yes

**Audience Explanation:**

The work is potentially of interest to some audience in TMLR, who work on feature selection, dimensionality reduction, model interpretability, among others. However, as detailed above, the framing of the work in RL is not convincing in the current form.

**Claims And Evidence:**

No

**Claims Explanation:**

The paper proposes a transfer-entropy-based method to select minimal state variables that are sufficient to learn a good policy. Under certain assumptions, the method provably finds such a minimal set. The paper also shows empirical support for the approach in both synthetic and popular RL benchmarks.

Reviewers have mixed evaluations. They appreciate the technical findings, such as the assumptions under which the proposed method will provably work, and the set of experiments that gives insights into how well it works. On the other hand, there are a few substantial weaknesses. First (and perhaps the most important) is the lack of strong motivation. The work requires learning a good policy upfront, then distilling a subset of state variables from the policy. Typically a smaller input would help speed up learning, but this is not the setting here. In the revision, the authors highlighted inference efficiency as the practical benefit, which still seems weak and hand-wavy. IMO the experiment should at least quantify the reduction of inference cost (or some proxy of it), and show the tradeoff with final policy quality (if any).

Second, the work relies on a few key assumptions. Reviewers are not convinced they are “weak”, while the revised paper still claims so in various places. If the authors believe the assumptions are weak, please justify it with further discussions and ideally empirical evidence in the experiments. If not, please be explicit --- it’s okay to rely on restricted assumptions as long as it’s made explicit and its limitations properly discussed.

We therefore recommend major revision as a resubmission.

**Resubmission Of Major Revision:**

The authors may consider submitting a major revision at a later time.